# The Effect of Optimal Self-Distillation in Noisy Gaussian Mixture Model

**Kaito Takanami**
Department of Physics
Graduate School of Science, The University of Tokyo,
Tokyo, Japan
Center for Interdisciplinary AI and Data Science, Ochanomizu University
Tokyo, Japan
takanami255@g.ecc.u-tokyo.ac.jp

**Takashi Takahashi**
Institute for Physics of Intelligence, The University of Tokyo
Tokyo, Japan
RIKEN center for AIP

**Ayaka Sakata**
Department of Information Science, Ochanomizu University
Tokyo, Japan
RIKEN center for AIP

## Abstract

Self-distillation (SD), a technique where a model improves itself using its own predictions, has attracted attention as a simple yet powerful approach in machine learning. Despite its widespread use, the mechanisms underlying its effectiveness remain unclear. In this study, we investigate the efficacy of hyperparameter-tuned multi-stage SD with a linear classifier for binary classification on noisy Gaussian mixture data. For the analysis, we employ the replica method from statistical physics. Our findings reveal that the primary driver of SD's performance improvement is denoising through hard pseudo-labels, namely discrete labels generated from the model's own predictions, with the most notable gains observed in moderately sized datasets. We also identify two practical heuristics to enhance SD: early stopping that limits the number of stages, which is broadly effective, and bias parameter fixing, which helps under label imbalance. To empirically validate our theoretical findings derived from our toy model, we conduct additional experiments on CIFAR-10 classification using pretrained ResNet backbone. These results provide both theoretical and practical insights, advancing our understanding and application of SD in noisy settings.

## 1 Introduction

Knowledge distillation (KD) Hinton et al. [2015] is a technique in machine learning that transfers the learned information from a complex model (often referred to as the teacher) to a simpler model (the student). This method attracted attention for achieving model compression with minimal performance loss, and has been applied across various domains, including image classification Liu et al. [2018], Xu et al. [2020], object detection Chen et al. [2017], and natural language processing Calderon et al. [2023], Gu et al. [2024].

39th Conference on Neural Information Processing Systems (NeurIPS 2025).

Among the various forms of KD, *self-distillation* (SD), originally termed *born again neural network* Furlanello et al. [2018] is particularly intriguing. In SD, the teacher and student models share identical architectures. This means that SD does not attempt the model compression; rather, it retrains the student model using the teacher's output. SD presents a intriguing paradox: despite training an identical model on the same dataset, the student model can outperform the teacher Furlanello et al. [2018], Hahn and Choi [2019], Clark et al. [2019], Yang et al. [2024], Chen et al. [2025].

Two main hypotheses have been proposed to explain such seemingly puzzling performance gains. The first suggests that the soft labels generated by the teacher provide *dark knowledge* Hinton et al. [2015]. Here, dark knowledge refers to the information implicitly embedded in the prediction probability distribution of the teacher model's output, which is absent in hard labels. It provides the student with additional information that captures subtle relationships within the data. The second hypothesis attributes the improvement to a denoising effect Das and Sanghavi [2023], Das et al. [2025] where the teacher model reduces the influence of the incorrect noisy labels in the training data, enabling the student model to learn a more reliable representation of the underlying patterns Pareek et al. [2024].

Although these hypotheses offer plausible explanations, the optimal behavior of SD, achieved through hyperparameter optimization and repeated iterations Pareek et al. [2024], remains poorly understood. This lack of understanding makes it difficult to identify the key factors that genuinely contribute to the performance improvement of SD. One reason for this difficulty is that exhaustive exploration of the hyperparameter space is usually computationally expensive, limiting the scope of experimental studies. As a result, evaluating the effectiveness of SD and identifying optimal strategies for its application remains a challenge.

To address this issue, we consider a multi-stage SD procedure using a linear classifier on Gaussian mixture data with label noise. This setup provides a controlled environment for analyzing both the dark knowledge and denoising hypotheses within a unified theoretical framework. In particular, we analyze this setting in the proportional asymptotic limit, where the input dimension $N$ and the data size $M$ diverge at the same rate, i.e., $N, M \to \infty$ with $M/N \to \alpha \in (0, \infty)$. A salient feature of this proportional asymptotic regime is that it allows precise characterization of the trained classifier's behavior, rather than merely providing rough lower or upper bounds. This enables us to explicitly determine optimal hyperparameters and iteration procedures, at least within simplified settings. Moreover, because this precise characterization involves only a finite number of variables, an exhaustive search for the optimal hyperparameters becomes computationally feasible. In this context, Gaussian mixture classification with linear models has served as a standard setting for gaining valuable insights into high-dimensional learning problems Dobriban and Wager [2018], Mignacco et al. [2020], Kini and Thrampoulidis [2020], Loureiro et al. [2021], Deng et al. [2022], Mannelli et al. [2024], Pesce et al. [2023], Takahashi [2024]. Technical tools for analyzing such asymptotic regimes include the replica method Mezard et al. [1986], Charbonneau et al. [2023], Convex Gaussian Min-max Theorem Thrampoulidis et al. [2015], Approximate Message Passing Donoho et al. [2009], which builds on Gordon's inequality Gordon [1988]. In our study of multi-stage SD, we employ the replica method, which has recently been shown to be applicable to multistage optimization problems, including self-training Takahashi [2022] and alternating minimization Okajima and Takahashi [2025].

Our main results are as follows:

1. The statistical properties of the trained classifiers are precisely characterized in the asymptotic limit where the input dimension and the data size diverge at the same rate. The precise formula for the generalization error is also be derived (Section 4).

2. SD using soft labels with dark knowledge can outperform hard-label training, particularly under low-noise or limited-data conditions. However, the performance gains achieved with soft labels are often quantitatively comparable to those obtained using hard labels across all settings we investigated. These findings suggest that, at least within our toy model, dark knowledge is not the key factor driving the success of SD (Section 5).

3. Naively applying multi-stage SD over too many stages degrades performance. This can be mitigated by employing an early-stopping heuristic that terminates the SD process at an appropriate stage. The resulting performance improvement is most pronounced for medium-sized datasets, where the denoising effect of SD is strongest. In addition, even though pseudo-labels contain label noise, performance comparable to training with ground-truth labels can be achieved in large-scale datasets (Section 6).

4. When ground-truth labels are imbalanced, learning solely from the teacher's pseudo-labels becomes challenging. This difficulty arises because the optimal regularization strength for aligning decision boundaries differs from that required for appropriately estimating the bias term. It is shown that fixing the bias term in the early stages of multi-stage SD serves as an effective heuristic to mitigate this issue (Section 7).

5. Experiments on CIFAR-10 with a pre-trained ResNet backbone qualitatively validate several theoretical predictions, extending beyond toy-model settings (Section 8).

These results provide a comprehensive understanding of the mechanisms underlying SD with a linear classifier on noisy Gaussian mixture data, and offer insights into how to optimally apply SD.

**Reproducibility:** The codes to reproduce some of our results are available at `https://github.com/taka255/self-distillation-analysis`.

**Impact statement:** We believe this work, which is a theoretical study of the learning behavior of simple linear model in a synthetic setting, does not have notable societal consequences.

## 2   Related Work

**Replica method for multi-stage learning.** The application of the replica method to analyzing the dynamics of high-dimensional systems was originally proposed for studying discrete optimization Krzakala and Kurchan [2007] and stochastic processes in glassy systems Franz and Parisi [2013]. In recent years, it has been extended to learning problems, particularly for analyzing sequential optimization processes, including self-training in semi-supervised learning Takahashi [2022] and alternating minimization Okajima and Takahashi [2025]. Our work builds on and advances this approach to analyze modern machine learning algorithms.

This methodology can be interpreted as a variant of Dynamical Mean Field Theory, a fundamental tool of statistical physics for analyzing the dynamics of high-dimensional systems, including gradient-based learning dynamics of neural networks Zou and Huang [2024], Helias and Dahmen [2020] (see appendix A for details).

**Theoretical analysis of self-distillation.** Theoretical analyses of distillation have predominantly focused on separable datasets due to their analytical tractability Phuong and Lampert [2019], Das et al. [2025], Das and Sanghavi [2023]. In separable settings, pseudo-labels generated by a teacher may be able to, in principle, match the ground-truth labels exactly. However, in realistic scenarios where data are not perfectly separable, pseudo-labels inevitably include errors. This fundamental limitation is not captured in these existing theoretical analyses, which rely on the separability assumption. While some exceptions Ji and Zhu [2020], Saglietti and Zdeborova [2022] have extended the analysis to non-separable datasets, they fall short of characterizing the behavior of optimal distillation under label noise. Our study fills this gap by quantitatively analyzing the improvements through SD with hyperparameter optimization on noisy and non-separable datasets.

## 3   Notations and Problem Setup

For convenience, we summarize all symbols and their definitions in Table B.1 of Appendix B, and provide a graphical illustration of the multi-stage SD model in Figure 6 of the same appendix.

### 3.1   Gaussian Mixture Data with Noisy Labels

We consider the binary classification of Gaussian mixture data with noisy labels using a single-layer neural network. Let $\boldsymbol{x}_\mu \in \mathbb{R}^N$ be the input data, where $\mu = 1, \ldots, M$ is the index of the data point and $N$ is the dimension of the input data. Here, we define the data-to-dimension ratio as $\alpha = M/N$. The true labels $y_\mu^{\text{true}} \in \{0, 1\}$ are independently generated according to the Bernoulli distribution $p(y_\mu^{\text{true}}) = \rho^{y_\mu^{\text{true}}} (1-\rho)^{1-y_\mu^{\text{true}}}$, with $\rho \in (0, 0.5]$. We consider a noisy observation in which the observed labels $y_\mu \in \{0, 1\}$ differ from the true labels with probability $\theta$: $\Pr[y_\mu \neq y_\mu^{\text{true}}] = \theta \in [0, 1/2]$. The feature vectors $\{\boldsymbol{x}_\mu\}_{1 \leq \mu \leq M}$ are generated from the Gaussian mixture distribution:

$$\boldsymbol{x}_\mu = (2y_\mu^{\text{true}} - 1)\boldsymbol{v}/\sqrt{N} + \sqrt{\Delta}\boldsymbol{z}_\mu, \tag{1}$$

where $\pm\boldsymbol{v} \in \mathbb{R}^N$ are the mean vectors of the Gaussian mixture, $\{\boldsymbol{z}_\mu\}_{1 \le \mu \le M}$ are i.i.d. standard Gaussian vectors, and $\Delta > 0$ controls the variance of the additive noise. [1]. Since the noise is rotation invariant, in the following, we set $\boldsymbol{v} = (1, 1, \ldots, 1)^\top$ without loss of generality. The goal is to train a good classifier from $D_{\mathrm{tr}} = \{\boldsymbol{x}_\mu, y_\mu\}_{\mu=1}^M$ that can classify an unseen observation $\boldsymbol{x}$, generated in the same way as in (1), correctly as $y^{\mathrm{true}}$.

### 3.2 Multi-stage Self-Distillation Model

We define the multi-stage SD model as a learning process that progresses through stages $t = 0, 1, 2, \ldots$. The loss function at the $t$-th stage is given by

$$\mathcal{L}_t(\boldsymbol{w}^t, B^t) = \sum_{\mu=1}^M \ell(y_\mu^t, Y(\boldsymbol{w}^t, B^t; \boldsymbol{x}_\mu)) + \frac{\lambda^t}{2} \left\| \boldsymbol{w}^t \right\|^2, \tag{2}$$

where $\ell(y, \hat{y})$ is a loss function, and the minimizer of Eq. (2) is denoted as $\hat{\boldsymbol{w}}^t$ and $\hat{B}^t$. Here, $Y(\boldsymbol{w}^t, B^t; \boldsymbol{x}_\mu)$ is the activation, and $y_\mu^t$ is the target label used for $t$-th stage learning. For $t = 0$, $y_\mu^t$ represents the observed label $y_\mu$, and for $t > 0$, it is interpreted as the pseudo-label. The activations and pseudo-labels are defined by the following rules.

**Activations:** The activation at the $t$-th stage is given by

$$Y(\boldsymbol{w}^t, B^t; \boldsymbol{x}_\mu) = \sigma\left( \frac{\boldsymbol{w}^t \cdot \boldsymbol{x}_\mu}{\sqrt{N}} + B^t \right), \tag{3}$$

where $\sigma(x)$ is an activation function, and the factor $1/\sqrt{N}$ ensures the output remains at order $\mathcal{O}(1)$ in $N$. We use two loss–activation combinations: (i) cross-entropy loss and sigmoid activation, $\ell(y, \hat{y}) = -y \log \hat{y} - (1 - y) \log(1 - \hat{y})$, $\sigma(x) = 1/(1 + \exp(-x))$; (ii) mean squared error loss and linear activation, $\ell(y, \hat{y}) = (y - \hat{y})^2$, $\sigma(x) = (x + 1)/2$. See Appendix B.3 for more details. These choices ensure convexity of the loss at each stage, which is crucial for our theoretical analysis. We refer to the model based on Eq. (2) as the $t$-SD model, with the 0-SD model as the base model before distillation. We refer to the $t$-SD model with cross-entropy loss as the *logistic $t$-SD model*, and the version with mean squared error loss as the *linear $t$-SD model*.

**Pseudo-Labels:** Labels at stage $t$ are computed from the $(t - 1)$-th stage as

$$y_\mu^t = \sigma\left( \beta^t \left( \frac{\hat{\boldsymbol{w}}^{t-1} \cdot \boldsymbol{x}^\mu}{\sqrt{N}} + \hat{B}^{t-1} \right) \right), \quad t \ge 1, \tag{4}$$

with $y_\mu^0 = y_\mu$. Here, $\beta^t > 0$ is the inverse temperature controlling the hardness of the pseudo-label. In the limit $\beta^t \to \infty$, it becomes hard (0 or 1), whereas finite $\beta^t$ yields soft labels in $(0, 1)$. Note that this parameter is meaningful only in the logistic SD model, since the linear SD model is scale-invariant.

### 3.3 Effect of Self-Distillation

We evaluate the effect of SD by measuring generalization error with optimal hyperparameters, tuned to minimize it. We define $\Theta^t = \{\lambda^i\}_{i=0}^t \cup \{\beta^j\}_{j=1}^t$ for $t \ge 0$, with $\Theta^0 = \{\lambda^0\}$, and define the error metrics:

$$\mathcal{E}^t = \mathbb{E}_\mathcal{D} \mathbb{E}_{\boldsymbol{x}, y^{\mathrm{true}}} \left[ \mathbb{I}\left( \hat{Y}(\hat{\boldsymbol{w}}^t, \hat{B}^t; \boldsymbol{x}) \ne y^{\mathrm{true}} \right) \right] \tag{5}$$

$$\mathcal{E}^{*t} = \min_{\Theta^t} \mathcal{E}^t, \quad \mathcal{E}^{*t}_{\mathrm{Hard}} = \min_{\lambda^0, \ldots, \lambda^t} \lim_{\beta^1, \cdots, \beta^t \to \infty} \mathcal{E}^t \quad (t \ge 1), \tag{6}$$

where $\hat{Y}(\hat{\boldsymbol{w}}^t, \hat{B}^t; \boldsymbol{x}) = \mathbb{I}[Y(\hat{\boldsymbol{w}}^t, \hat{B}^t; \boldsymbol{x}) > 1/2]$, $\mathbb{I}(A)$ is the indicator function, $\mathcal{D} = \{(\boldsymbol{x}_\mu, y_\mu^{\mathrm{true}}, y_\mu)\}_{1 \le \mu \le M}$, and $(\boldsymbol{x}, y^{\mathrm{true}})$ is a test input generated in the same way as the training data. $\mathcal{E}^{*t}_{\mathrm{Hard}}$ represents the error when dark knowledge is removed by hardening soft labels. This limit is meaningful only for the logistic $t$-SD model. We include this as a reference to assess the role of dark knowledge. A significantly smaller $\mathcal{E}^{*t}$ than $\mathcal{E}^{*0}$ indicates SD effectively improves generalization.

---

[1]The results remain valid if $\boldsymbol{z}_\mu$ are replaced by i.i.d. random vectors with zero mean, unit covariance, and finite higher-order moments due to the central limit theorem.

## 4 Precise characterization of multi-stage self-distillation

For a new input $\boldsymbol{x}$ generated from the Gaussian mixture as in (1), the distribution of the pre-activation $\hat{\boldsymbol{w}}^t \cdot \boldsymbol{x}/\sqrt{N} + \hat{B}^t$ can be characterized as

$$\frac{\hat{\boldsymbol{w}}^t \cdot \boldsymbol{x}}{\sqrt{N}} + \hat{B}^t \stackrel{\mathrm{d}}{=} \bar{m}^t(2y^{\mathrm{true}} - 1) + \hat{B}^t + \sqrt{\Delta \bar{Q}^{tt}} z, \tag{7}$$

where $\stackrel{\mathrm{d}}{=}$ denotes equality in distribution, $\bar{Q}^{tt} = \|\hat{\boldsymbol{w}}^t\|^2/N$ represents the norm of the weight vector, $\bar{m}^t = \hat{\boldsymbol{w}}^t \cdot \boldsymbol{v}/N$ is its alignment with the cluster center direction $\boldsymbol{v}$, and $z \sim \mathcal{N}(0,1)$. The label $y^{\mathrm{true}}$ follows $p(y^{\mathrm{true}}) = \rho^{y^{\mathrm{true}}}(1-\rho)^{1-y^{\mathrm{true}}}$. As such, the term $\bar{m}^t(2y^{\mathrm{true}} - 1)$ represents the signal component, while $\sqrt{\bar{Q}^{tt}}$ controls the uncertainty due to the variance in the classifier's weights. This result follows from the independence of $\boldsymbol{x}$ from $D_{\mathrm{tr}}$ and its Gaussianity.

At the proportional limit $N, M \to \infty, M/N \to \alpha \in (0, \infty)$, these quantities are expected to converge to deterministic values, which do not fluctuate aginst the realization of $D_{\mathrm{tr}}$, as

$$\bar{Q}^{tt}(D_{\mathrm{tr}}) \to Q^{tt}, \quad \bar{m}^t(D_{\mathrm{tr}}) \to m^t, \quad \hat{B}^t(D_{\mathrm{tr}}) \to b^t. \tag{8}$$

This leads to the following expression of the average generalization error Mignacco et al. [2020].

**Proposition 4.1.** *Under the proportional asymptotic limit ($N, M \to \infty$, constrained by $M/N \to \alpha \in (0, \infty)$), the average generalization error of the t-SD model is given by*

$$\mathcal{E}^t = \rho H\left(\frac{m^t + b^t}{\sqrt{\Delta Q^{tt}}}\right) + (1 - \rho)H\left(\frac{m^t - b^t}{\sqrt{\Delta Q^{tt}}}\right), \tag{9}$$

*where $H(x) = 1 - \int_{-\infty}^{x} \mathrm{d}t\, e^{-t^2/2}/\sqrt{2\pi}$.*

Proposition 4.1 indicates that the key quantities to evaluate the generalization error are the *alignment* $\bar{m}^t/\sqrt{\bar{Q}^{tt}}$ (the cosine similarity between the direction of the decision boundary and $\boldsymbol{v}$) and the *rescaled bias* $\hat{B}^t/\sqrt{\bar{Q}^{tt}}$ (the offset of the decision boundary from the origin).

Since $\bar{Q}^{tt}, \bar{m}^t$ and $\hat{B}^t$ are expected to be concentrated to deterministic values as in (8), they are evaluated by investigating the average values $\mathbb{E}_{\mathcal{D}}[\bar{Q}^{tt}], \mathbb{E}_{\mathcal{D}}[\bar{m}^t], \mathbb{E}_{\mathcal{D}}[\hat{B}^t]$ at the large system limit. For evaluating these averaged quantities, we used the replica method and obtained the following results, which precisely characterize the values of them.

**Result 1** (Statistics of the T-SD model parameters). *There exist constant matrices $\hat{Q} = (\hat{Q}^{st}), \hat{\chi} = (\hat{\chi}^{st}) \in \mathbb{R}^{(T+1) \times (T+1)}$ and a constant vector $\hat{m} = (\hat{m}^t) \in \mathbb{R}^{T+1}$ such that, in the proportional asymptotic limit ($N, M \to \infty, M/N \to \alpha \in (0, \infty)$),*

$$\hat{w}_i^0 \stackrel{\mathrm{d}}{=} \frac{1}{\hat{Q}^{00} + \lambda^0}\left(\hat{m}^0 + \hat{\xi}^0\right) \quad \text{and} \quad \hat{w}_i^T \stackrel{\mathrm{d}}{=} \frac{1}{\hat{Q}^{TT} + \lambda^T}\left(\hat{m}^T + \hat{\xi}^T - \sum_{s=0}^{T-1} \hat{Q}^{st}\hat{w}^s\right) \quad (T \geq 1), \tag{10}$$

*where $\hat{\boldsymbol{\xi}} = (\hat{\xi}^t) \in \mathbb{R}^{T+1} \sim \mathcal{N}(\mathbf{0}, \hat{\chi})$.*

We deliberately use the notation *Result* rather than *Theorem*, as the derivation relies on the replica method is not yet a mathematically rigorous proof.[2] The derivation is given in appendix C, where the explicit determination of the constant matrices $\hat{Q}, \hat{\chi}$ and the vector $\hat{m}$ is also detailed. Furthermore, Result 1 is validated by numerical simulations in appendix D, demonstrating good consistency.

The constants $\hat{Q}, \hat{m}$ and $\hat{\chi}$ that appear in Theorem 1 and $b^t$ in Equation 9 can all be obtained by solving at most $\mathcal{O}(T)$ coupled equations (see Appendix C), so they can be computed efficiently. Moreover, from Equation (10) one directly obtains $Q^{tt} = \mathbb{E}[(\hat{w}_i^t)^2]$ and $m^t = \mathbb{E}[\hat{w}_i^t]$. Together with Proposition 4.1, this gives an exact characterization of the generalization error in the asymptotic limit.

Beyond yielding a closed-form for the asymptotic generalization error, Theorem 1 also makes clear that the parameters learned via SD are effectively governed by only a small number of key order

---

[2]For possible directions toward rigorous proofs and the associated technical challenges, see the remark in appendix C.10.

parameters. To see how this plays out in practice, we now unpack the roles of the parameters in Theorem 1. The weights are composed of three components: (i) the signal term $\hat{m}^t$ representing the amount of correlation with the cluster center $\boldsymbol{v}$, (ii) the noise term $\hat{\xi}^t$ capturing the randomness inherent in the data, and (iii) the correction term $-\sum_{s=1}^{t-1} \hat{Q}^{st} \hat{w}^s$ accounting for correlations induced by the labels generated by previous teacher models, i.e. 0-th to $(t-1)$-th SD models.

## 5 The role of soft labels in self-distillation

In this section, we focus on the $t = 1$ case, where SD involves a single teacher and student.

**Dark knowledge effect is marginal.** We investigate the generalization error improvement of the optimal logistic 1-SD model in a noiseless setting ($\theta = 0$), where any improvement would stem solely from the teacher's soft labels, because there is no label noise to be removed. As shown in Figure 1, the linear 1-SD model exhibits only modest improvement across various dataset sizes and data variances. Gains are slightly more visible when the dataset is small and variance is low, but even under such conditions, the improvement remains under $0.4\%$. These results suggest that in linear models, the contribution of dark knowledge is limited.

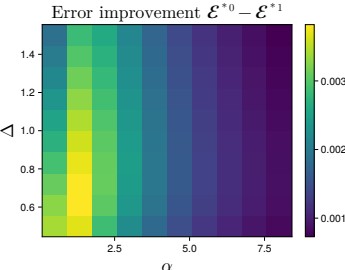

Figure 1: Heat map of the improvement error $\mathcal{E}^{*0} - \mathcal{E}^{*1}$ at $\rho = 0.4$ and $\theta = 0$ in linear 1-SD model.

**Soft labels vs. hard labels.** Soft labels provide dark knowledge beyond simple class predictions. However, since denoising primarily aims to identify the correct label, this additional information may not always be necessary. Motivated by the limited effect of dark knowledge observed in the noiseless case, we hypothesize that hard labels, obtained by binarizing soft labels, may suffice for SD. To test this hypothesis, we compare generalization errors in logistic 1-SD using hard and soft labels.

Figure 2A and B show the improvement in generalization errors achieved using soft labels ($\mathcal{E}^{*0} - \mathcal{E}^{*1}$) and hard labels ($\mathcal{E}^{*0} - \mathcal{E}^{*1}_{\text{Hard}}$), respectively. Both exhibit similar qualitative trends: large improvements are observed in higher noise and larger dataset settings, consistent with the denoising interpretation.

Figure 2C shows the ratio of the improvements in A and B, revealing two distinct regions. In the dark purple region, soft labels offer a clear advantage over hard labels, indicating that dark knowledge contributes meaningfully to SD. In contrast, the bright yellow region shows that hard labels are nearly as effective as soft labels, suggesting that dark knowledge is unnecessary in these conditions. However, even in the region where soft labels outperform hard labels, the quantitative improvement is small (Figure 2A). These findings suggest that while soft labels may be beneficial in specific conditions, their overall impact on SD performance is relatively limited in our setting.

This observation may refine our understanding of dark knowledge in SD. While previous studies Ma et al. [2022], Mandal et al. [2025] emphasized its general importance, our results suggest that its effectiveness depends on dataset characteristics and noise level, and that it is not always necessary.

## 6 Understanding the effect of multi-stages

So far, we have examined the role of dark knowledge using the logistic $t$-SD model in single-stage settings. Here, we shift our focus and consider the effect of repeated SD across multiple stages.

To investigate this, we consider the $t$-stage model ($t \geq 2$) and study how performance evolves with repeated applications of SD. Due to computational constraints, we focus on the linear $t$-SD model. Although it does not distinguish between soft and hard labels, we expect it to still capture the essential

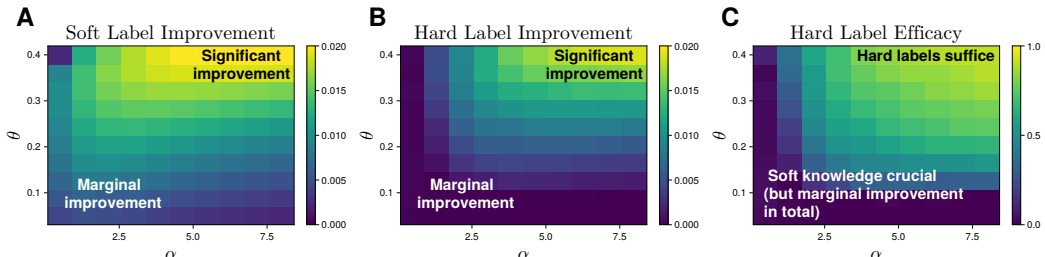

Figure 2: (A) and (B): generalization error improvements in the optimal logistic 1-SD model using soft labels ($\mathcal{E}^{*0} - \mathcal{E}^{*1}$) and hard labels ($\mathcal{E}^{*0} - \mathcal{E}^{*1}_{\text{Hard}}$), respectively. (C): the ratio of the two: $(\mathcal{E}^{*0} - \mathcal{E}^{*1}_{\text{Hard}})/(\mathcal{E}^{*0} - \mathcal{E}^{*1})$. Parameters: $\rho = 0.4$, $\Delta = 1.0$.

behavior of multi-stage SD. We first focus on the label-balanced case ($\rho = 0.5$), and discuss the label-imbalanced setting ($\rho < 0.5$) in Section 7.

**Denoising effects and dataset size dependence.** We investigated the denoising effect of SD by comparing the optimal $t$-SD model with two baselines: the optimal 0-SD model ($\mathcal{E}^{*0}$ with $\theta > 0$) and the optimal 0-SD model trained on noiseless data ($\mathcal{E}^{*0}$ with $\theta = 0$), as shown in Figure 3A.

The behavior of $\mathcal{E}^{*t}$ can be categorized into three types depending on $\alpha$: large $\alpha$, intermediate $\alpha$, and small $\alpha$. When $\alpha$ is sufficiently large ($\alpha \gtrsim 10$) or small ($\alpha \lesssim 0.2$), the decrease in $\mathcal{E}^{*t}$ slows down around $t = 3$, with $\mathcal{E}^{*3}$ and $\mathcal{E}^{*10}$ being almost the same, as shown in Figure 3A.

At sufficiently large $\alpha$ ($\alpha \gtrsim 10^1$ in Figure 3A), multi-stage SD quickly achieves performance close to that of noise-free setting. Specifically, by $t = 3$, the generalization error $\mathcal{E}^{*t}$ becomes nearly equal to the noise-free case $\mathcal{E}^{*0}$ with $\theta = 0$, even though the error before distillation $\mathcal{E}^{*0}$ with $\theta > 0$ is clearly higher than noise-free case. This finding is noteworthy, as perfect noise correction is challenging due to the overlapping data distributions. In contrast, for small $\alpha$, $\mathcal{E}^{*t}$ remains close to $\mathcal{E}^{*0}$ even at $t = 10$, indicating that SD's denoising effect is limited. Unlike large or small $\alpha$, in the intermediate range, the improvement progresses slowly with the number of iterations, but $\mathcal{E}^{*t}$ approaches to the noise-free case as $t$ increases. As a consequence, the error improvement by multi-stage SD exhibits a non-monotonic behavior with dataset size, with the most pronounced denoising effect observed for moderately-sized datasets, as indicated by the arrow in Figure 3A.

Intuitively, this phenomenon can be explained as follows. For small datasets, the limited data makes it difficult for the teacher to learn pseudo-labels that effectively correct label noise, reducing the effectiveness of SD. In contrast, for large datasets, the teacher becomes strong enough to generate pseudo-labels that allow SD to correct noise effectively. However, the pre-SD classifier already performs well, leaving little room for further improvement. For intermediate dataset sizes, the available data is sufficient for the teacher to produce pseudo-labels that enable noise correction. At the same time, the pre-SD classifier remains suboptimal. As a result, the most substantial performance gains are observed in this intermediate $\alpha$ region.

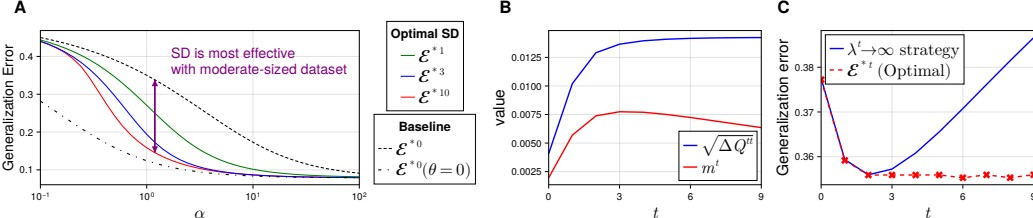

Figure 3: (A) A comparison of the optimal generalization error for the linear $t$-SD model, 0-SD model, and the noiseless case. (B) Dynamics of $\sqrt{\Delta Q^{tt}}$ and $m^t$ for the linear $t$-SD model with $\lambda^0, \cdots, \lambda^t \to \infty$. (C) Comparison of generalization error between the linear $t$-SD model with $\lambda^0, \cdots, \lambda^t \to \infty$ and optimal $t$-SD. Parameters for (A): $\rho = 0.5, \Delta = 0.5, \theta = 0.4$. (B, C): $\alpha = 1.0, \Delta = 1.2, \theta = 0.3, \beta^t = 1/\sqrt{Q^{t-1,t-1}}$.

**Fixed point analysis and learning dynamics.** A natural question may be how the error of $t$-SD behaves as $t \to \infty$. To investigate this, the limit $\lambda^0, \ldots, \lambda^t \to \infty$ is considered, under which a closed-form expression for the generalization error of the linear $t$-SD model can be derived. This provides theoretical predictions for the error as $t \to \infty$. This setting is not only mathematically tractable but also interpretable: it simplifies the solution to a the averaging estimator $\hat{\boldsymbol{w}}^t \propto \sum_{\mu=1}^{M} y_\mu \boldsymbol{x}_\mu$. As shown in Dobriban and Wager [2018], Lelarge and Miolane [2019], Mignacco et al. [2020], this averaging estimator is Bayes-optimal when noise-less ($\theta = 0$) and balanced ($\rho = 1/2$) case. Furthermore, Akhtiamov et al. [2024] shows that in the $\lambda \to \infty$ regime, any corruption rate ($\theta < 0.5$) in the balanced case ($\rho = 1/2$) can be eliminated. We also confirmed in preliminary experiments that the optimal hyperparameter schedule obtained by black-box optimization starts with very large regularization and then switches to zero, which naturally supports early stopping as a sensible strategy. Based on the above considerations, we now present the following theorem for this large regularization limit.

**Result 2** (**The generalization error at** $t \to \infty$)**.** *For an arbitrary choice of the set of the temperature parameters* $\{\beta^t\}_{t \geq 0}$*, the generalization error of the linear $t$-SD model with $\rho = 0.5$, $\lambda^0, \cdots, \lambda^t \to \infty$ and $t \to \infty$ is given by* $\lim_{t \to \infty} \mathcal{E}^t = 0.5$ *whenever* $\alpha < \Delta^2$*.*

The proof is given in appendix F.

Result 2 shows that, under certain data conditions, naively continuing multi-stage SD can reduce the model's performance to the level of random guessing. In particular, when $\alpha = \Delta^2$, the generalization error exhibits a phase transition that separates meaningful predictors from random ones (see Appendix F).

To gain deeper insights into the learning dynamics, we analyze the time evolution of $m^t = \mathbb{E}[\hat{\boldsymbol{w}}^t \cdot \boldsymbol{v}]/N$ and $Q^{tt} = \mathbb{E}[\hat{\boldsymbol{w}}^t \cdot \hat{\boldsymbol{w}}^t]/N$, which quantify signal extraction and prediction uncertainty, respectively. The temperature is set to $\beta^t = 1/\sqrt{Q^{t-1,t-1}}$ to prevent the norm of the weight vector, $\|\hat{\boldsymbol{w}}^t\|$, from vanishing. The values of $m^t$ and $Q^{tt}$ are plotted in Figure 3B. As shown in the figure, $m^t$ peaks during the initial iterations, whereas the predictive uncertainty $\sqrt{\Delta Q^{tt}}$ increases steadily throughout the stages, leading to performance decline. Therefore, optimal learning may be achieved by halting the training process when signal extraction is maximized. Interestingly, this early stopping strategy closely matches the results obtained through comprehensive hyperparameter optimization across the entire model (Figure 3C).

These results are consistent with experimental studies Zhang and Sabuncu [2020], where the term "diversity" specifically refers to the predictive uncertainty of teacher predictions, which has been suggested to relate to the success of SD. Our result may support that such predictive uncertainty (diversity) plays a key role in effective signal extraction. However, the results also imply that the extractable signal saturates after a few iterations, highlighting an intrinsic limit to the benefit of repeated distillation.

## 7 The hardness of learning bias in label imbalanced cases

Next, we examine the label imbalanced case $\rho < 0.5$, where performance improvement results from the interplay between alignment and rescaled bias, and compare it to the $\rho = 0.5$ case.

The difference from the label-balanced case lies in the difficulty of simultaneously learning the bias and the alignment in imbalanced datasets. Figure 4B shows the evolution of the rescaled bias ($b^t/\sqrt{Q^{tt}}$) and alignment ($m^t/\sqrt{Q^{tt}}$) over distillation stages. As the figure indicates, while alignment improves gradually, the rescaled bias worsens as training progresses, deviating from its Bayes-optimal value.

This behavior can be attributed to the effect of ridge regularization, which acts only on the weight vector $\boldsymbol{w}^t$. Strong regularization, which may be necessary to improve alignment, shrinks the norm the weight ($Q^{tt} = \|\hat{\boldsymbol{w}}^t\|_2^2/N$) and consequently increases the rescaled bias. When the rescaled bias becomes too large compared to the alignment, the model tends to classify most data into a single class (either positive or negative), resulting in poor generalization performance (see also Eq. (9)).

Hence, in label-imbalanced cases, loss minimization may not be suitable for jointly optimizing bias and weight. In contrast, for balanced data, the optimal bias is simply zero ($b^t = 0$), and no such trade-off arises.

To address the challenge of balancing bias and alignment, we find that fixing the bias at an early stage is a simple and effective heuristic. Similar approaches, which separate the training of alignment and bias, have also been proposed in logistic regression Mignacco et al. [2020] and self-training Takahashi [2022] in imbalanced Gaussian mixtures. The dotted lines in Figure 4B illustrate the results with the bias fixed at its value obtained in the optimal 0-SD, followed by performing the optimal $t$-SD. Figure 4C further compares the generalization error of $t$-SD with and without this heuristic. Applying bias fixing significantly improves both rescaled bias and alignment, and exhibits convergence towards the Bayes-optimal solution, as observed in the $\rho = 0.5$ case.

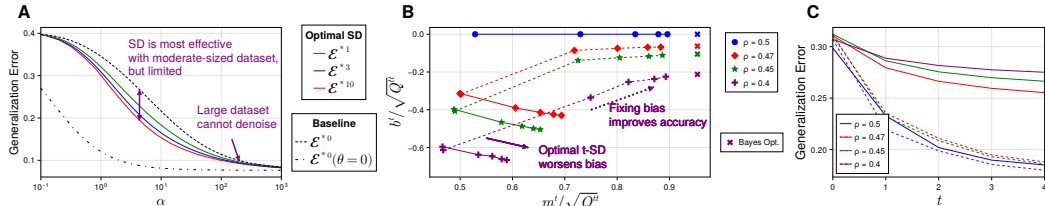

Figure 4: (A) Optimal generalization error of the linear $t$-SD model compared with the 0-SD model and the noiseless case under label imbalance ($\rho = 0.4$). (B) Evolution of the rescaled bias ($|b^t|/\sqrt{Q^{tt}}$) and alignment ($m^t/\sqrt{Q^{tt}}$) from $t = 0$ to $t = 4$ for the optimal $t$-SD model (solid lines) and the variant with fixed bias (dotted lines). (C) Generalization error over stages $t$ for the same models as in (B). Parameters: (A) $\rho = 0.4$, $\Delta = 0.5$, $\theta = 0.4$; (B, C) $\Delta = 1.0$, $\theta = 0.4$, $\alpha = 10.0$.

## 8    Experiments on real datasets

We have analyzed the behavior of multi-stage SD using the Gaussian mixture model with label noise, which allowed precise asymptotic characterization and provided valuable insights. However, this setting is highly idealized, and there remains a significant gap between this toy model and real-world datasets. To bridge the gap, we conduct a sanity check to test whether our theoretical predictions hold in a standard vision task. We fine-tune only the final layer of a ResNet pretrained on IMAGENET1K_V2 maintainers and contributors [2016] (BSD 3-Clause "New" License) with $L_2$ regularization on noisy CIFAR-10 (cat vs. dog) Krizhevsky et al. [2009] (MIT License) and compare the results to our theoretical predictions. See Appendix G for experimental details.

Figure 5 shows the generalization error of optimal 1-SD, compared to optimal 0-SD and optimal 1-SD using hard labels (see Eq. 6). When using ResNet-18 for feature extraction, we observe virtually no improvement due to SD in the large-$\alpha$ region, while in the middle-$\alpha$ region SD achieves a denoising effect that yields $\sim 5\%$ performance gain. According to the results in Section 6, the benefit of SD should peak at moderate $\alpha$ and decline again as $\alpha$ becomes even smaller; however, this downturn cannot be observed here because it corresponds to an excessively small sample size in our setup. Nevertheless, Figure 5 partially reproduce the prediction from Section 6 that SD's effectiveness is maximized at intermediate values of $\alpha$.

Next, focusing on the improvement in generalization error when propagating hard labels from 0-SD to 1-SD, we find that the relative importance of soft-label error reduction grows as $\alpha$ decreases, mirroring the results of Section 5. Remarkably, as anticipated in Section 5, even hard labels, which completely discard dark knowledge, still provide a significant denoising benefit. We observe a similar trend when using ResNet-50; however, since this model has a higher baseline performance, the overall magnitude of improvement is smaller. See also Appendix H for further results.

## 9    Conclusion

We investigated optimal multi-stage SD with a linear classifier for binary classification on noisy Gaussian mixture data. The technical crux of our analysis is a precise asymptotic formula for multi-stage SD, derived using the replica method from statistical physics. As this formula involves only a small, finite number of variables, it enables exhaustive hyperparameter search at a reasonable computational cost. This contrasts with experimental studies, where exhaustive exploration of the hyperparameter space is often computationally intractable. By using this, we obtained the following

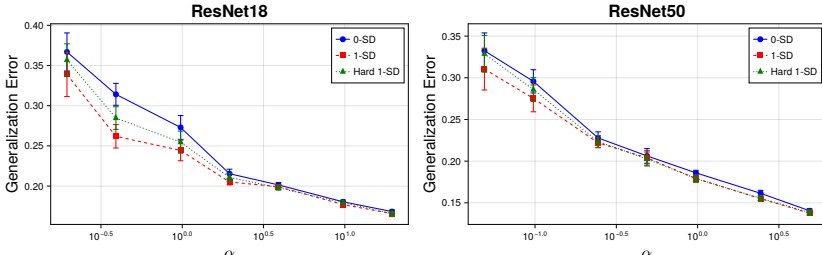

Figure 5: Comparison of the optimal generalization error of the logistic 0-SD model, 1-SD model and 1-SD model using hard pseudo labels for CIFAR-10 dog versus cat classification using pretrained ResNet-18 ($N = 512$) and ResNet-50 ($N = 2048$) feature representations. Parameters: $\theta = 0.4$. Error bars represent the standard error of the mean over 10 trials per point.

results. First, we found that dark knowledge in soft-labels plays a more limited role than previously assumed, with denoising likely being the primary driver of its success. SD's strong denoising capability is evident even with inseparable data distributions. Second, SD is most effective with moderate dataset sizes, showing weaker effects in both very small datasets (where denoising is difficult) and very large datasets (where noise has small impact). Third, fixing the bias and focusing on alignment optimization serves as a useful heuristic in SD. More broadly, this suggests a general strategy for multi-stage SD: progressively narrowing the parameters optimized at each stage. These findings not only enhance our theoretical understanding of SD mechanisms but also provide a foundation for developing improved algorithms.

**Limitations:** While our asymptotic analysis provides valuable insights, it is limited to linear models under a Gaussian mixture setting. Extending the analysis to deep networks or alternative SD strategies remains an important direction for future work. See Appendix I for further discussion.

## Acknowledgements

The study was conducted as part of the exploratory project "Mathematical Exploration of Universal Structures in Multicomponent and Polydisperse Systems," supported by the Toyota Konpon Research Institute, Inc. This work was also supported by JSPS KAKENHI Grant Numbers 22H05117 and 23K16960, JST ACT-X Grant Number JPMJAX24CG, and JST BOOST NAIS Grant Number JPMJBS2418.

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

# A Further remarks on related works

**Replica method for machine learning problems.** As machine learning models and datasets grow increasingly complex, traditional mathematical approaches often fall short in providing rigorous analytical solutions. This complexity gap has led to a rising demand for alternative theoretical tools that can offer insights into model behavior and performance, even when rigorous mathematical solutions are out of reach.

In this context, the replica method, originally developed in statistical physics, has emerged as a powerful analytical technique for machine learning problems. While not yet mathematically rigorous in all aspects, this method has been widely applied to various models, from simple perceptrons Gardner [1988] to modern non-i.i.d. datasets Gerace et al. [2020], Loureiro et al. [2022], with some of its predictions later rigorously proven Barbier et al. [2019]. The replica method offers unique advantages, such as the ability to compute exact generalization errors rather than bounds or necessary conditions. This precision enables explicit optimization of hyperparameters in multi-stage SD, providing deeper insights into model behavior and performance.

**Relationship between multi-stage replica method and DMFT.** Traditional DMFT Helias and Dahmen [2020] is primarily used for analyzing learning dynamics. This approach is effective when the system's state at time $t$ can be expressed explicitly using the state at time $t-1$, allowing for direct averaging over data. However, in more complex scenarios like SD, conventional DMFT techniques face challenges. In our model, the transition from one state to the next is not explicitly defined but is implicitly determined through an optimization process. Specifically, the previous state $(\hat{\boldsymbol{w}}^{t-1}, \hat{B}^{t-1})$ influences the output labels $y^t$ (Eq. (4)), which then feed into the optimization problem minimizing the loss function (Eq. (2)) to determine the new state $(\hat{\boldsymbol{w}}^t, \hat{B}^t)$. This implicit dependency, mediated by an optimization step, makes the dynamics more complex than those typically handled by traditional DMFT approaches. To overcome these challenges and extend DMFT's applicability to such complex scenarios, we employ the replica method, which allows us to analyze these implicit optimization-based state transitions effectively.

# B   Further remarks on model and notation

In this appendix, we compile and summarize all notation used throughout the paper, provide detailed commentary on the model's structure, and include illustrative figures to aid the reader's understanding.

## B.1   Notation

Table B.1 lists each symbol and its definition.

Table 1: Summary of Notations

| Category | Symbol | Definition |
|---|---|---|
| *Data Generation* | $N$ | Dimension of input features |
| | $M$ | Number of training samples |
| | $\alpha = M/N$ | Data-to-dimension ratio, $\mathcal{O}(1)$ |
| | $\boldsymbol{x}_\mu$ | Input vector ($\mu = 1, \ldots, M$) |
| | $\rho$ | Class prior ($\rho \in (0, 0.5]$) |
| | $y_\mu^{\text{true}}$ | True label, $\{0, 1\}$ with $p(y^{\text{true}}) = \rho^y (1-\rho)^{1-y}$ |
| | $y_\mu$ | Noisy observed label; $\Pr[y_\mu \neq y_\mu^{\text{true}}] = \theta$ |
| | $\theta$ | Label noise rate ($\theta \in (0, 0.5]$) |
| | $\boldsymbol{v}$ | Gaussian mean vector, set $(1, 1, \ldots, 1)^\top$ w.l.o.g. |
| | $\Delta$ | Feature noise variance in (1), $\mathcal{O}(1)$ |
| | $D_{\text{tr}}$ | Training set $\{(\boldsymbol{x}_\mu, y_\mu)\}_{\mu=1}^M$ |
| *Distillation Process* | $t$ | Stage index (0 = base model) |
| | $(\boldsymbol{w}^t, B^t)$ | Weights and bias at stage $t$ (minimize (2)) |
| | $y_\mu^t$ | Target label at stage $t$: $y_\mu^0 = y_\mu$, for $t > 0$ see (4) |
| | $\lambda^t$ | $L_2$-regularization strength in (2) |
| | $\beta^t$ | Inverse temperature for soft pseudo-label hardness in (4) |
| *Loss & Prediction* | $\ell(y, \hat{y})$ | Loss function: {cross-entropy, MSE} |
| | $\sigma(x)$ | Activation: sigmoid $1/(1 + e^{-x})$ or linear $(x+1)/2$ |
| | $Y(\boldsymbol{w}^t, B^t; \boldsymbol{x})$ | Activation output, see Eq. (3) |
| | $\mathcal{L}_t(\boldsymbol{w}^t, B^t)$ | Stage-$t$ objective, see Eq. (2) |
| *Performance Metrics* | $\mathcal{E}^t$ | Generalization error after stage $t$, def. in Sec. 3 |
| | $\mathcal{E}^{*0}$ | Optimal 0-SD error (minimized over $\lambda^0$) |
| | $\mathcal{E}^{*t}$ | Optimal $t$-SD error (minimized over $\lambda^{0..t}, \beta^{1..t}$) |
| | $\mathcal{E}_{\text{Hard}}^{*t}$ | Optimal $t$-SD error ($\beta^{1..t} \to \infty$ and minimized over $\lambda^{0..t}$) |
| *Asymptotic Quantities* | $Q^{st}$ | Weight–weight overlap, limit of $\hat{w}^s \cdot \hat{w}^t / N$ |
| | $m^t$ | Weight–signal overlap, $\hat{w}^t \cdot v / N$ |
| | $b^t$ | Rescaled bias, $\hat{B}^t / \sqrt{Q^{tt}}$ |

## B.2   Illustration of the model

We present a graphical illustration of the model. Figure 6 depicts the overall architecture, highlighting its key components and the flow of information.

## B.3   Motivation for the choice of activation function

We chose the activation function $\sigma(x) = (x + 1)/2$ instead of the simpler $\sigma(x) = x$ because it ensures that the decision boundary remains unchanged when adjusting the temperature parameter. This choice isolates the effects of soft labels, avoiding confounding influences from shifting decision boundaries.

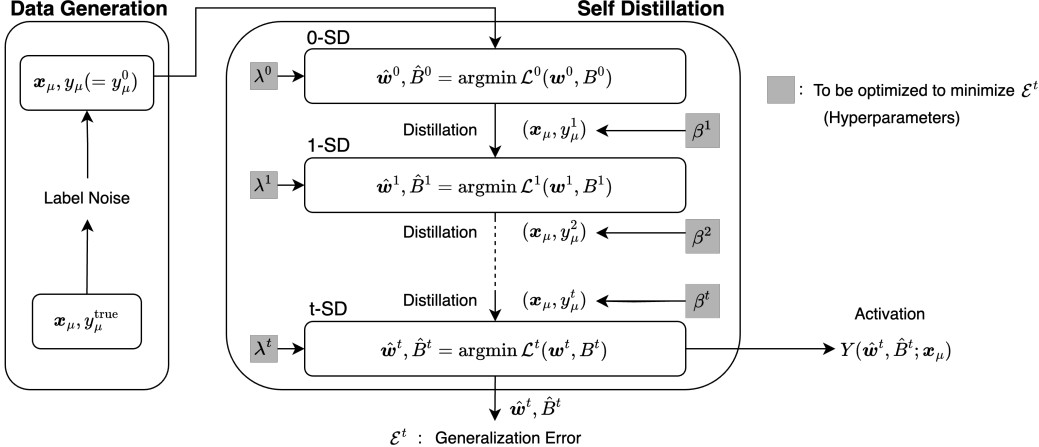

Figure 6: A schematic diagram of the $t$-SD model.

## C    Replica calculation

This appendix outlines the procedure for deriving Result 1 using the replica method. The process can be summarized in the following steps:

- Present the complete statements of the results, including Results 1 (Subsection C.1).

- Introduce the average of the $p$-th moment of $\hat{\boldsymbol{w}}^1, \cdots, \hat{\boldsymbol{w}}^t$, denoted as $\mathcal{F}_p^t$, which characterizes macroscopic quantities such as generalization error (Subsection C.2).

- Show that the joint probability distribution of $\hat{\boldsymbol{w}}^t$ and $\hat{B}^t$ coincides with the correlation function of a system obtained by duplicating the original system, referred to as the replica system (Subsection C.3 for $t = 0$ and Subsection C.4 for $t > 0$).

- Outline the procedure for evaluating $\mathcal{F}_p^t$ by incorporating certain assumptions into the replica variables (Subsections C.5 and C.6).

- Derive equations to determine the parameters necessary for calculating $\mathcal{F}_p^t$ (Subsections C.7 and C.8).

- Use these equations to derive Result C.1 (Subsection C.9).

Finally, we present remarks on the rigorous proofs in Section C.10.

### C.1    Full Statement of the Result

Here we give the complete statements of the results to be proved.

**Theorem C.1.** *(**Statistics of the T-SD model**) Under the proportional asymptotic limit ($N, M \to \infty$, constrained by $M/N \to \alpha \in (0, \infty)$), we have*

$$\hat{w}_i^0 \overset{\mathrm{d}}{=} \frac{1}{\hat{Q}^{00} + \lambda^0} \left( \hat{m}^0 + \hat{\xi}^0 \right) \tag{11}$$

$$\hat{w}_i^T \overset{\mathrm{d}}{=} \frac{1}{\hat{Q}^{TT} + \lambda^T} \left( \hat{m}^T + \hat{\xi}^T - \sum_{s=0}^{T-1} \hat{Q}^{st} \hat{w}^s \right) \quad (T \geq 1) \tag{12}$$

$$\frac{\hat{\boldsymbol{w}}^T \cdot \boldsymbol{x}_\mu}{\sqrt{N}} + \hat{B}^T \overset{\mathrm{d}}{=} h^T + z_*^T, \tag{13}$$

*where the parameters satisfy the following equations:*

$$
\begin{cases}
Q^{0t} & = \frac{\hat{m}^0 m^t + R^{0t}}{\lambda^0 + \hat{Q}^{00}} \\
Q^{st} & = \frac{\hat{m}^s m^t + R^{st} - \sum_{l=0}^{s-1} \hat{Q}^{ls} Q^{lt}}{\lambda^t + \hat{Q}^{tt}} \quad (t \geq s \geq 1)
\end{cases}
\tag{14}
$$

$$
\begin{cases}
R^{s0} & = \frac{\hat{\chi}^{0s}}{\hat{Q}^{00} + \lambda^0} \\
R^{st} & = \frac{\hat{\chi}^{st} - \sum_{l=0}^{t-1} \hat{Q}^{lt} R^{sl}}{\hat{Q}^{tt} + \lambda^t} \quad (t \geq 1)
\end{cases}
\tag{15}
$$

$$
\begin{cases}
m^0 & = \frac{\hat{m}^0}{\lambda^0 + \hat{Q}^{00}} \\
m^t & = \frac{\hat{m}^t - \sum_{s=0}^{t-1} \hat{Q}^{st} m^s}{\lambda^t + \hat{Q}^{tt}} \quad (t \geq 1)
\end{cases}
\tag{16}
$$

$$
\begin{cases}
\chi^{ss} & = \frac{1}{\lambda^s + \hat{Q}^{ss}} \\
\chi^{s,t+1} & = -\frac{\hat{Q}^{t,t+1}}{\lambda^{t+1} + \hat{Q}^{t+1,t+1}} \chi^{st} \quad (t \geq s)
\end{cases}
\tag{17}
$$

$$
\begin{cases}
\hat{Q}^{st} & = -\frac{\alpha}{\chi^{tt}} \mathbb{E}_{y, y^{\mathrm{true}}, \boldsymbol{\xi}} \left[ \frac{\mathrm{d} z_*^t}{\mathrm{d} h^s} \right] \\
\hat{m}^t & = \frac{\alpha}{\Delta \chi^{tt}} \mathbb{E}_{y, y^{\mathrm{true}}, \boldsymbol{\xi}} \left[ (2y - 1) z_*^t \right] \\
\hat{\chi}^{st} & = \frac{\alpha}{\Delta \chi^{ss} \chi^{tt}} \mathbb{E}_{y, y^{\mathrm{true}}, \boldsymbol{\xi}} \left[ z_*^s z_*^t \right] \\
\mathbb{E}_{y, y^{\mathrm{true}}, \boldsymbol{\xi}} \left[ z_*^t \right] & = 0.
\end{cases}
\tag{18}
$$

*Here, $Q = (Q^{st})$, $\chi = (\chi^{st})$ and $\hat{\chi} = (\hat{\chi}^{st})$ are symmetric matrices in $\mathbb{R}^{(T+1) \times (T+1)}$, and we introduced the following notations:*

$$
z_*^t = \underset{z^t}{\arg\min} \left[ \frac{(z^t)^2}{2 \Delta \chi^{tt}} + \ell \left( y^t, \sigma \left( h^t + z^t \right) \right) \right]
\tag{19}
$$

$$
\begin{cases}
y^0 & = y \\
y^t & = \sigma \left( \beta^{t-1} \left( h^{t-1} + z_*^{t-1} \right) \right) \quad (t \geq 1)
\end{cases}
\tag{20}
$$

$$
\begin{cases}
h^0 & \overset{\mathrm{d}}{=} \xi^0 + (2 y^{\mathrm{true}} - 1) m^0 + b^0 \\
h^t & \overset{\mathrm{d}}{=} \xi^t + \sum_{s=1}^{t-1} \frac{\chi^{st}}{\chi^{ss}} z_*^s + (2 y^{\mathrm{true}} - 1) m^t + b^t \quad (t \geq 1),
\end{cases}
\tag{21}
$$

*where $\boldsymbol{\xi} = (\xi^t) \in \mathbb{R}^{T+1} \sim \mathcal{N}(\mathbf{0}, \Delta Q)$, $\hat{\boldsymbol{\xi}} = (\hat{\xi}^t) \in \mathbb{R}^{T+1} \sim \mathcal{N}(\mathbf{0}, \hat{\chi})$, and $y, y^{\mathrm{true}} \in \{0, 1\}$ that are generated as $p(y, y^{\mathrm{true}}) = p(y \mid y^{\mathrm{true}}) p(y^{\mathrm{true}})$ with $p(y^{\mathrm{true}}) = \rho^{y^{\mathrm{true}}} (1 - \rho)^{1 - y^{\mathrm{true}}}, p(y \neq y^{\mathrm{true}} \mid y^{\mathrm{true}}) = \theta$.*

## C.2 What to calculate

Our primary interest lies in understanding how macroscopic quantities, such as the generalization error, behave under fluctuations in the training data. These macroscopic quantities can generally be expressed, excluding the bias, as functions of

$$
\frac{1}{N} \sum_i (\hat{w}_i^0)^{p^0} \cdots (\hat{w}_i^t)^{p^t},
\tag{22}
$$

where $p^0, \cdots, p^t \in \mathbb{N} \cup 0$. In the asymptotic limit $N \to \infty$, we expect that Eq. (22) converges with probability 1 to their expected values

$$
\lim_{N \to \infty} \frac{1}{N} \sum_i (w_i^0)^{p^0} \cdots (w_i^t)^{p^t} = \lim_{N \to \infty} \frac{1}{N} \mathbb{E}_{\mathcal{D}} \left[ \sum_i (w_i^0)^{p^0} \cdots (w_i^t)^{p^t} \right], \quad \text{almost surely.}
\tag{23}
$$

This kind of concentration is called the self-averaging property Mezard et al. [1986] in the context of statistical mechanics. Although it is not obvious that the self-averaging property holds, this property has been proved in several convex optimization problems Thrampoulidis et al. [2015].

Thus, the quantity we need to compute is

$$
\mathcal{F}_{\boldsymbol{p}}^t = \lim_{N \to \infty} \mathbb{E}_{\mathcal{D}} \left[ (\hat{w}_i^0)^{p^0} \cdots (\hat{w}_i^t)^{p^t} \right],
\tag{24}
$$

where $\boldsymbol{p} = (p^0, \cdots, p^t)$.

Since the optimization problem for each time step $t$ in our model is convex, $\hat{\boldsymbol{w}}^t$ and $\hat{b}^t$ are deterministically determined as $(\hat{\boldsymbol{w}}^0, \hat{B}^0) \to (\hat{\boldsymbol{w}}^1, \hat{B}^1) \to \cdots \to (\hat{\boldsymbol{w}}^t, \hat{B}^t)$ given the data $\mathcal{D} = \{\boldsymbol{x}_\mu, y_\mu^{\text{true}}, y_\mu\}_\mu$. However, to facilitate our analysis, we intentionally treat this deterministic process as a stochastic one. Specifically, we model the transition from $(\hat{\boldsymbol{w}}^{s-1}, \hat{B}^{s-1})$ to $(\hat{\boldsymbol{w}}^s, \hat{B}^s)$ using the following distribution

$$\hat{\boldsymbol{w}}^s, \hat{B}^s \sim p(\boldsymbol{w}^s, B^s \mid \hat{\boldsymbol{w}}^{s-1}, \hat{B}^{s-1}, \mathcal{D}) = \lim_{\gamma^s \to \infty} \frac{\exp(-\gamma^s \mathcal{L}_s(\boldsymbol{w}^s, B^s))}{Z^s} \quad (s = 1, \cdots, t), \quad (25)$$

where $\mathcal{L}_s$ is defined in Eq. (2) and

$$Z^s = \int \mathrm{d}\boldsymbol{w}^s \, \mathrm{d}B^s \, \exp(-\gamma^s \mathcal{L}_s(\boldsymbol{w}^s, B^s)) \quad (s = 1, \cdots, t) \quad (26)$$

is the marginal likelihood (partition function). Observe that in the limit as $\gamma^s \to \infty$, the distribution concentrates on $\operatorname{argmin}_{\boldsymbol{w}^s, B^s} \mathcal{L}_s(\boldsymbol{w}^s, B^s)$. Similary, the case $t = 0$ is defined as

$$\hat{\boldsymbol{w}}^0, \hat{B}^0 \sim p(\boldsymbol{w}^0, B^0 \mid \mathcal{D}) = \lim_{\gamma^0 \to \infty} \frac{\exp(-\gamma^0 \mathcal{L}_0(\boldsymbol{w}^0, B^0))}{Z^0} \quad (s = 1, \cdots, t), \quad (27)$$

where

$$Z^0 = \int \mathrm{d}\boldsymbol{w}^0 \, \mathrm{d}B^0 \, \exp(-\gamma^0 \mathcal{L}_0(\boldsymbol{w}^0, B^0)). \quad (28)$$

Following the probabilistic interpretation of these dynamics, the quantity defined in Eq. (24) can be reformulated as

$$\mathcal{F}_{\boldsymbol{p}}^t = \mathbb{E}_{\mathcal{D}} \left[ \left\langle \cdots \left\langle \langle w_i^t \rangle_t^{p^t} \left( w_i^{t-1} \right)^{p^{t-1}} \right\rangle_{t-1} \cdots \left( w_i^0 \right)^{p^0} \right\rangle_0 \right], \quad (29)$$

where $\langle f(w^s) \rangle_s$ is expectation under the distribution $p(\boldsymbol{w}^s, B^s \mid \boldsymbol{w}^{s-1}, B^{s-1}, \mathcal{D})$ if $s > 0$ and $p(\boldsymbol{w}^0, B^0 \mid \mathcal{D})$ if $s = 0$.

In summary, our computational task is to calculate the data average of statistical quantities $\mathcal{F}_{\boldsymbol{p}}^t$ (Eq. (29)) for a sequence of random variables $(\hat{\boldsymbol{w}}^0, \hat{B}^0) \to (\hat{\boldsymbol{w}}^1, \hat{B}^1) \to \cdots \to (\hat{\boldsymbol{w}}^t, \hat{B}^t)$ following a Markov process defined by Eqs. (25) and (27).

### C.3 One-stage replica method

To grasp the outline of the calculation in the replica method, we first consider the case of $t = 0$. For simplicity of notation, in the calculations within Subsections C.3 and C.4, we treat $\boldsymbol{w}^t$ as one-dimensional, omitting the subscript $i$ from $w_i^t$. However, it is straightforward to extend this to the general $N$-dimensional case.

The data average of the $p$-th moment of the solution $\hat{w}^0$ (Eq. (27)) is given by

$$\mathcal{F}_p^0 = \mathbb{E}_{\mathcal{D}}[(\hat{w}^0)^p] = \mathbb{E}_{\mathcal{D}} \left[ \int \mathrm{d}w^0 \, \mathrm{d}B^0 \, w^0 p(w^0, B^0 \mid \mathcal{D}) \right]^p \quad (30)$$

$$= \lim_{\gamma^0 \to \infty} \mathbb{E}_{\mathcal{D}} \left[ \int \mathrm{d}w^0 \, \mathrm{d}B^0 \, w^0 \frac{\exp(-\gamma^0 \mathcal{L}_0(w^0, B^0))}{Z^0} \right]^p. \quad (31)$$

Direct computation of this is challenging due to the presence of the marginal likelihood $Z^0$ in the denominator of Eq. (31). To circumvent this difficulty, we introduce the following identity that holds for any $p \in \mathbb{N}$:

$$\mathbb{E}_{\mathcal{D}}[(\hat{w}^0)^p] = \lim_{n^0 \to 0} \lim_{\gamma^0 \to \infty} \frac{\mathcal{W}_p(n^0, \gamma^0)}{\Xi_p(n^0, \gamma^0)} \quad (32)$$

where

$$\mathcal{W}_p(n^0, \gamma^0) = \mathrm{E}_\mathcal{D}\left[\left\{\int \mathrm{d}w^0\, \mathrm{d}B^0\, w^0 \exp(-\gamma^0 \mathcal{L}(w^0, B^0))\right\}^p (Z^0)^{n^0-p}\right], \tag{33}$$

$$\Xi(n^0, \gamma^0) = \mathrm{E}_\mathcal{D}[(Z^0)^{n^0}]. \tag{34}$$

For the expectation with respect to data, we resort to a calculation method known as replica method. First, we assume that $n^t \in \mathbb{N}$ and $n^t > p$,[3] and express (33) and (34) by using $n^t$-replicated variables $w_1^t, \ldots, w_{n^t}^t$ as

$$\mathcal{W}_p(n^0, \gamma^0) = \mathrm{E}_\mathcal{D}\left[\int \mathrm{d}\boldsymbol{w}^0\, \mathrm{d}\boldsymbol{B}^0\, w_1^0 \cdots w_p^0 \exp\left(-\gamma^0 \sum_{a_0=1}^{n_0} \mathcal{L}(w_{a_0}^0, B_{a_0}^0)\right)\right] \tag{35}$$

$$= \int \mathrm{d}\boldsymbol{w}^0\, \mathrm{d}\boldsymbol{B}^0\, w_1^0 \cdots w_p^0\, \mathrm{E}_\mathcal{D}\left[\exp\left(-\gamma^0 \sum_{a_0=1}^{n_0} \mathcal{L}(w_{a_0}^0, B_{a_0}^0)\right)\right] \tag{36}$$

$$\Xi(n^0, \gamma^0) = \mathrm{E}_\mathcal{D}\left[\int \mathrm{d}\boldsymbol{w}^0\, \mathrm{d}\boldsymbol{B}^0\, \exp\left(-\gamma^0 \sum_{a_0=1}^{n_0} \mathcal{L}(w_{a_0}^0, B_{a_0}^0)\right)\right] \tag{37}$$

$$= \int \mathrm{d}\boldsymbol{w}^0\, \mathrm{d}\boldsymbol{B}^0\, \mathrm{E}_\mathcal{D}\left[\exp\left(-\gamma^0 \sum_{a_0=1}^{n_0} \mathcal{L}(w_{a_0}^0, B_{a_0}^0)\right)\right], \tag{38}$$

where $\boldsymbol{w}^0$ and $\boldsymbol{B}^0$ are shorthands for $w_1^0, \cdots, w_{n^0}^0$ and $B_1^0, \cdots, B_{n^0}^0$, respectively. These expression indicates that $\mathcal{W}_p(n^0, \gamma^0)/\Xi(n^0, \gamma^0)$ can be regarded as the expectation of the $p$-body correlation of replica variables obeying the joint distribution

$$p(\boldsymbol{w}^0, \boldsymbol{B}^0) = \lim_{\gamma^0 \to \infty} \frac{1}{\Xi(n^0, \gamma^0)} \mathrm{E}_\mathcal{D}\left[\exp\left(-\gamma^0 \sum_{a_0=1}^{n^0} \mathcal{L}(w_{a_0}^0, B_{a_0}^0)\right)\right]. \tag{39}$$

The primary challenge in our initial calculations was the necessity of averaging over the data, which significantly complicated the process. However, by employing the replica method, as shown in Eq. (39), we effectively incorporate the data average into the probability distribution of the variables, thereby enabling us to derive their statistical properties. The system that follows the probability distribution given by Eq. (39) is called a replica system.

## C.4 Extention to multi-stage replica method

While the preceding analysis follows the conventional replica method prescription, the current scenario presents a unique challenge: the dependence of each estimator $\hat{w}^t$ on its predecessor $\hat{w}^{t-1}$ significantly increases the complexity of the problem. To address this issue, we employ an innovative approach that involves the recursive application of the replica trick at each stage of the process.

To illustrate this approach, let us consider how Eq. (39) evolves along step $t$ under our recursive methodology. First we define the following recursive function:

$$f_{t-1}(w^{t-1}) = \left\langle w^t \right\rangle_t^{p^t} \tag{40}$$

$$f_s(w^s) = \left\langle f_{s+1}(w^{s+1})\left(w^{s+1}\right)^{p^{s+1}} \right\rangle_{s+1} \quad (0 \le s < t-1). \tag{41}$$

Then, what we want to calculate (Eq. (29)) is expressed as

$$\mathcal{F}_{\boldsymbol{p}}^t = \mathbb{E}_\mathcal{D}\left\langle f_0(w^0)(w^0)^{p^0} \right\rangle_0. \tag{42}$$

---

[3]This assumption is not consistent with taking the limit $n^0 \to 0$, as performed in Eq. (32). Therefore, it remains necessary to verify whether the results obtained under the condition $n^0 > p$ can be correctly extrapolated to the regime where $n^0 \to 0$. While the mathematical validity of this analytic continuation has not yet been rigorously proven, no counterexamples to its validity have been identified so far, at least in cases where the optimization problem determining the parameters $\hat{w}^t$ is convex.

On the other hand, assuming that $n^0, \cdots, n^t \in \mathbb{N}$, $n^t > p$ and $n^0, \cdots, n^{t-1} > 1$, one can deduce

$$f_{t-1}(w^{t-1}) = \left\langle w^t \right\rangle_t^{p^t} \tag{43}$$

$$= \left\{ \int \mathrm{d}w^t \, \mathrm{d}B^t \, w^t \exp\left(-\gamma^t \mathcal{L}_t(w^t, B^t)\right) \right\}^{p^t} \left(Z^t\right)^{n^t - p^t} \tag{44}$$

$$= \int \mathrm{d}\boldsymbol{w}^t \, \mathrm{d}\boldsymbol{B}^t \, w_1^t \cdots w_{p^t}^t \exp\left(-\gamma^t \sum_{a_t=1}^{n^t} \mathcal{L}_t(w_{a_t}^t, B_{a_t}^t)\right) \tag{45}$$

$$f_{s-1}(w^{s-1}) = \left\langle f_s(w^s)(w^s)^{p^s} \right\rangle_s \tag{46}$$

$$= \left\{ \int \mathrm{d}w^s \, \mathrm{d}B^s \, f_s(w^s)(w^s)^{p^s} \exp\left(-\gamma^s \mathcal{L}_s(w^s, B^s)\right) \right\}^{p^s} \left(Z^s\right)^{n^s - p^s} \tag{47}$$

$$= \int \mathrm{d}\boldsymbol{w}^s \, \mathrm{d}\boldsymbol{B}^s \, f_s(w_1^s) w_1^s \cdots w_{p^s}^s \exp\left(-\gamma^s \sum_{a_s=1}^{n^s} \mathcal{L}_s(w_{a_s}^s, B_{a_s}^s)\right) \quad (1 \le s < t) \tag{48}$$

$$\mathcal{F}_{\boldsymbol{p}}^t = \left\langle f_0(w^0)(w^0)^{p^0} \right\rangle_0 \tag{49}$$

$$= \left\{ \int \mathrm{d}w^0 \, \mathrm{d}B^0 \, f_0(w^0)(w^0)^{p^0} \exp\left(-\gamma^0 \mathcal{L}_s(w^0, B^0)\right) \right\}^{p^0} \left(Z^0\right)^{n^0 - p^0} \tag{50}$$

$$= \int \mathrm{d}\boldsymbol{w}^0 \, \mathrm{d}\boldsymbol{B}^0 \, f_0(w_1^0) w_1^0 \cdots w_{p^0}^0 \exp\left(-\gamma^0 \sum_{a_0=1}^{n^0} \mathcal{L}_0(w_{a_0}^0, B_{a_0}^0)\right), \tag{51}$$

where $\boldsymbol{w}^t$ and $\boldsymbol{B}^t$ are shorthands for $w_1^t, \cdots, w_{n^t}^t$ and $B_1^t, \cdots, B_{n^t}^t$, respectively. These equations demonstrate that when considering the parameters at time s while keeping the parameters at time $s - 1$ fixed, the distribution of replica variables $\boldsymbol{w}^s$ and $\boldsymbol{B}^s$ at time $s$ depends on the first replica variables $w_1^{s-1}$ and $b_1^{s-1}$ at time $s - 1$. By recursively substituting Eqs. (45) and (48) into (51), we have

$$\mathcal{F}_{\boldsymbol{p}}^t = \lim_{\substack{\gamma^0 \to \infty \\ n^0 \to 0}} \cdots \lim_{\substack{\gamma^t \to \infty \\ n^t \to 0}} \frac{\mathcal{W}_{\boldsymbol{p}}^t(\boldsymbol{n}, \boldsymbol{\gamma})}{\Xi^t(\boldsymbol{n}, \boldsymbol{\gamma})}, \tag{52}$$

where $\boldsymbol{n} = (n^0, \cdots, n^t)$, $\boldsymbol{\gamma} = (\gamma^0, \cdots, \gamma^t)$ and

$$\mathcal{W}_{\boldsymbol{p}}^t(\boldsymbol{n}, \boldsymbol{\gamma}) = \int \mathrm{d}\boldsymbol{w}^0 \cdots \mathrm{d}\boldsymbol{w}^t \, \mathrm{d}\boldsymbol{B}^0 \cdots \mathrm{d}\boldsymbol{B}^t \left((w_1^t \cdots w_{p^t}^t) \times (w_1^{t-1})^{p^{t-1}} \times \cdots \times (w_1^0)^{p^0}\right) \tag{53}$$

$$\times \mathbb{E}_{\mathcal{D}} \left[\exp\left(-\sum_{s=1}^t \gamma^s \sum_{a_s=1}^{n_s} \tilde{\mathcal{L}}_s(w_{a^s}^s, B_{a^s}^s)\right)\right] \tag{54}$$

$$\Xi^t(\boldsymbol{n}, \boldsymbol{\gamma}) = \mathbb{E}_{\mathcal{D}}\left[(Z^t)^{n^t} \times (Z^{t-1})^{n^{t-1}} \times \cdots \times (Z^0)^{n^0}\right] \tag{55}$$

$$= \mathbb{E}_{\mathcal{D}} \left[\int \mathrm{d}\boldsymbol{w}^0 \cdots \mathrm{d}\boldsymbol{w}^t \, \mathrm{d}\boldsymbol{B}^0 \cdots \mathrm{d}\boldsymbol{B}^t \exp\left(-\sum_{s=1}^t \gamma^s \sum_{a_s=1}^{n_s} \tilde{\mathcal{L}}_s(w_{a^s}^s, B_{a^s}^s)\right)\right] \tag{56}$$

$$= \int \mathrm{d}\boldsymbol{w}^0 \cdots \mathrm{d}\boldsymbol{w}^t \, \mathrm{d}\boldsymbol{B}^0 \cdots \mathrm{d}\boldsymbol{B}^t \, \mathbb{E}_{\mathcal{D}} \left[\exp\left(-\sum_{s=1}^t \gamma^s \sum_{a_s=1}^{n_s} \tilde{\mathcal{L}}_s(w_{a^s}^s, B_{a^s}^s)\right)\right], \tag{57}$$

where $\tilde{\mathcal{L}}(w_{a_t}^t, B_{a^t}^t)$ represents the loss when the first replica at $t-1$ is used as a pseudo-label for training, i.e.,

$$\tilde{\mathcal{L}}(w_{a_t}^t, B_{a^t}^t) = \sum_\mu \ell(\tilde{y}_\mu^t, Y(w_{a_t}^t, B_{a^t}^t; x_\mu)) + \frac{\lambda^t}{2} \|w_{a_t}^t\|^2 \tag{58}$$

$$\tilde{y}_\mu^t = \sigma\left(\beta^t\left(\frac{\hat{w}_1^{t-1} \cdot x_\mu + \hat{B}_1^{t-1}}{\sqrt{N}}\right)\right) \quad (t > 0) \tag{59}$$

$$\tilde{y}_\mu^0 = y_\mu. \tag{60}$$

These expression indicates that $\mathcal{W}_p^t(\boldsymbol{n}, \boldsymbol{\gamma})/\Xi^t(\boldsymbol{n}, \boldsymbol{\gamma})$ can be regarded as the expectation of the $(p^0 + \cdots + p^t)$-body correlation of replica variables obeying the joint distribution

$$p(\boldsymbol{w}^0, \cdots, \boldsymbol{w}^t, \boldsymbol{B}^0, \cdots, \boldsymbol{B}^t) = \lim_{\gamma^0 \cdots \gamma^t \to \infty} \frac{1}{\Xi^t(\boldsymbol{n}, \boldsymbol{\gamma})} \mathrm{E}_\mathcal{D}\left[\exp\left(-\sum_{s=1}^t \gamma^s \sum_{a_s=1}^{n_s} \tilde{\mathcal{L}}_s(w_{a^s}^s, B_{a^s}^s)\right)\right]. \tag{61}$$

Ultimately, our problem reduces to calculating the statistical properties of the replica variables $\boldsymbol{w}^0, \cdots, \boldsymbol{w}^t$ that follow the distribution given by Eq. (61). Following the standard prescription for analysis in the asymptotic limit, it is crucial to investigate the behavior of the replica partition function Eq. (57) for this analysis.

### C.5   The calculation of the replica partition function

From this subsection, we recover the subscript of dimension $i$ in $w_{a_t, i}^t$. Our next step is to calculate data average in the replica partition function (Eq. (57)). To achieve this, we first calculate the partition function for a finite $n^1, \cdots, n^t$, and then consider the limit as $n^1, \cdots, n^t \to 0$.

First, we define the linearly transformed variable $\boldsymbol{u}^t = (u_1^t, \cdots, u_{a_t}^t, \cdots, u_{n^t}^t)^T$ as

$$\boldsymbol{u}^t = \sqrt{\frac{\Delta}{N}} \sum_i \boldsymbol{w}_i^t z_i, \tag{62}$$

where $z_i$ is standard normal random variavle defined in Eq. (1) and $\boldsymbol{w}_i^t = (w_{1,i}^t, \cdots, w_{a_t,i}^t, \cdots, w_{n^t,i}^t)^T$. Then, $\boldsymbol{u} = ((\boldsymbol{u}^1)^T, \cdots, (\boldsymbol{u}^T)^T)$ also follows a Gaussian distribution, with the mean and covariance given by

$$\mathbb{E}_\mathcal{D}[u_{a_t}^t] = 0, \quad \mathbb{E}_\mathcal{D}[u_{a_s}^s u_{c_t}^t] = \Delta Q_{a_s c_t}^{st}, \tag{63}$$

where $Q_{a_s c_t}^{st}$ is defined as

$$Q_{a_s c_t}^{st} = \frac{1}{N} \boldsymbol{w}_{a_s}^s \cdot \boldsymbol{w}_{c_t}^t. \tag{64}$$

Then, the partition function of the replica distribution Eq. (57) can be expressed as

$$\Xi^T(\boldsymbol{n}, \boldsymbol{\gamma}) = \int \boldsymbol{w}^0 \cdots \boldsymbol{w}^T \int \mathrm{d}\boldsymbol{B}^0 \cdots \boldsymbol{B}^T \exp\left(-\sum_t \frac{\lambda^t}{2} \sum_{a_t} \|\boldsymbol{w}_{a_t}^t\|^2\right)$$

$$\times \prod_{\mu=1}^M \mathbb{E}_{\boldsymbol{x}_\mu, y_\mu, y_\mu^{\text{true}}} \exp\left[-\sum_t \sum_{a_t} \gamma^t \ell\left(\tilde{y}_\mu^t, \sigma\left(\frac{\boldsymbol{w}_{a_t}^t \cdot \boldsymbol{x}_\mu}{\sqrt{N}} + B_{a_t}^t\right)\right)\right] \qquad (65)$$

$$= \int \mathrm{d}\boldsymbol{Q} \, \mathrm{d}\boldsymbol{m} \int \boldsymbol{w}^0 \cdots \boldsymbol{w}^T \int \mathrm{d}\boldsymbol{B}^0 \cdots \boldsymbol{B}^T \left[\prod_{st} \prod_{a_s, c_t} \delta\left(NQ_{a_s c_t}^{st} - \boldsymbol{w}_{a_s}^s \boldsymbol{w}_{c_t}^t\right)\right]$$

$$\times \left[\prod_t \prod_{a_t} \delta\left(Nm_{a_t}^t - \boldsymbol{w}_{a_t}^t \cdot \boldsymbol{v}\right)\right]$$

$$\times \prod_{i=1}^N \exp\left(-\sum_t \frac{\lambda^t}{2} \sum_{a_t} |w_{a_t,i}^t|^2\right)$$

$$\times \prod_{\mu=1}^M \mathbb{E}_{\boldsymbol{x}_\mu, y_\mu, y_\mu^{\text{true}}} \exp\left[-\sum_t \sum_{a_t} \gamma^t \ell\left(\tilde{y}_\mu^t, \sigma\left(\frac{\boldsymbol{w}_{a_t}^t \cdot \boldsymbol{x}_\mu}{\sqrt{N}} + B_{a_t}^t\right)\right)\right], \qquad (66)$$

where $\int \mathrm{d}\boldsymbol{Q}$ is an integration over $\{Q_{a_s c_t}^{st}\}_{1 \leq s \leq t \leq T, 1 \leq a_s \leq n^s, 1 \leq c_t \leq n^t}$ and $\int \mathrm{d}\boldsymbol{m}$ is an integration over $\{m_{a_t}^t\}_{1 \leq t \leq T, 1 \leq a_t \leq n^t}$. Using the following integral representations of the Dirac delta function[4]:

$$\delta\left(NQ_{a_s c_t}^{st} - \boldsymbol{w}_{a_s}^s \cdot \boldsymbol{w}_{c_t}^t\right) = \int \mathrm{d}\hat{Q}_{a_s c_t}^{st} \exp\left(-\frac{\hat{Q}_{a_s c_t}^{st}}{2}\left(NQ_{a_s c_t}^{st} - \boldsymbol{w}_{a_s}^s \cdot \boldsymbol{w}_{c_t}^t\right)\right) \qquad (67)$$

$$\delta(Nm_{t_a}^t - \boldsymbol{v} \cdot \boldsymbol{w}_{a_t}^t) = \int \mathrm{d}\hat{m}_{a_t}^t \exp\left(-\hat{m}_{a_t}^t(Nm_{a_t}^t - \boldsymbol{v} \cdot \boldsymbol{w}_{a_t}^t)\right). \qquad (68)$$

One can find that Eq. (66) can be separated into three terms: (1) an interaction term $G_I$, which shows the interaction between order parameters (parameters without hat) and conjugate parameters (parameters with hat); (2) an entropic term $G_S$, which scales as $N$; and (3) an energy term $G_E$, which scales as $M$. Based on these observations, calculating each component of the partition function yields the following equation:

$$\Xi^T(\boldsymbol{n}, \boldsymbol{\gamma}) = \int \mathrm{d}\boldsymbol{Q} \, \mathrm{d}\boldsymbol{m} \, \mathrm{d}\hat{\boldsymbol{Q}} \, \mathrm{d}\hat{\boldsymbol{m}} \, \mathrm{d}\boldsymbol{B}^0 \cdots \mathrm{d}\boldsymbol{B}^T \, (G_I)^N (G_S)^N (G_E)^M \qquad (69)$$

where

$$G_I = \exp\left[-\left(\frac{1}{2}\sum_{st}\sum_{a_s c_t} \hat{Q}_{a_s c_t}^{st} Q_{a_s c_t}^{st} + \sum_t \sum_{a_t} \hat{m}_{a_t} m_{a_t}\right)\right] \qquad (70)$$

$$G_S = \int \mathrm{d}\boldsymbol{w}^0 \cdots \boldsymbol{w}^T \exp\left[-\sum_t \sum_a \frac{\gamma^t}{2}\lambda^t(w_a^t)^2 + \sum_t \sum_a \hat{m}_{a_t}^t w_{a_t}^t + \frac{1}{2}\sum_{st}\sum_{a_s c_t} \hat{Q}_{a_s c_t}^{st} w_{a_s}^s w_{c_t}^t\right] \qquad (71)$$

$$G_E = \mathbb{E}_{y^{\text{true}}, y} \mathbb{E}_{\boldsymbol{u}} \exp\left[-\gamma^0 \sum_{a_0} \ell\left(y, \sigma\left((2y^{\text{true}} - 1)m_{a_0}^0 + u_{a_0}^0 + B_{a_0}^0\right)\right)\right]$$

$$\times \prod_{t=1}^T \exp\left[-\gamma^t \sum_{a_t} \ell\left(\sigma\left((2y^{\text{true}} - 1)m_1^{t-1} + u_1^{t-1} + B_1^{t-1}\right), \sigma\left((2y^{\text{true}} - 1)m_{a_t}^t + u_{a_t}^t + B_{a_t}^t\right)\right)\right]. \qquad (72)$$

---

[4]The integrations in Eqs. (67) and (68) are performed along the imaginary axis.

In the asimptotic limit ($N, M \to \infty, \alpha = M/N = \mathcal{O}(1)$), Eq. (69) can evaluated using the saddle point method. Using this technique, the partition function (Eq. (69)) is evaluated as

$$\Xi^T(\boldsymbol{n}, \boldsymbol{\gamma}) = \exp\left[N \max\left[\Psi(\boldsymbol{Q}, \boldsymbol{m}, \hat{\boldsymbol{Q}}, \hat{\boldsymbol{m}}, \boldsymbol{B}^0, \cdots, \boldsymbol{B}^T)\right]\right] \tag{73}$$

$$= \exp\left[N\left[\Psi(\boldsymbol{Q}^*, \boldsymbol{m}^*, \hat{\boldsymbol{Q}}^*, \hat{\boldsymbol{m}}^*, \boldsymbol{B}^{0*}, \cdots, \boldsymbol{B}^{T*})\right]\right], \tag{74}$$

where

$$\Psi(\boldsymbol{Q}, \boldsymbol{m}, \hat{\boldsymbol{Q}}, \hat{\boldsymbol{m}}, \boldsymbol{B}^0, \cdots, \boldsymbol{B}^T) = \log G_I + \log G_S + \alpha \log G_E. \tag{75}$$

and

$$\boldsymbol{Q}^*, \boldsymbol{m}^*, \hat{\boldsymbol{Q}}^*, \hat{\boldsymbol{m}}^*, \boldsymbol{B}^{0*}, \cdots, \boldsymbol{B}^{T*} = \operatorname*{argmax}_{\boldsymbol{Q}, \boldsymbol{m}, \hat{\boldsymbol{Q}}, \hat{\boldsymbol{m}}, \boldsymbol{B}^0, \cdots, \boldsymbol{B}^T} \Psi(\boldsymbol{Q}, \boldsymbol{m}, \hat{\boldsymbol{Q}}, \hat{\boldsymbol{m}}, \boldsymbol{B}^0, \cdots, \boldsymbol{B}^T). \tag{76}$$

The saddle point equations to minimize $\Psi(\boldsymbol{Q}, \boldsymbol{m}, \hat{\boldsymbol{Q}}, \hat{\boldsymbol{m}}, \boldsymbol{B}^0, \cdots, \boldsymbol{B}^T)$ are given by

$$Q^{st}_{a_s c_t} = \frac{1}{G_S}\frac{\partial G_S}{\partial \hat{Q}^{st}_{a_s c_t}}, \quad m^t_{a_t} = \frac{1}{G_S}\frac{\partial G_S}{\partial \hat{m}^t_{a_t}}, \quad \hat{Q}^{st}_{a_s c_t} = \alpha\frac{1}{G_E}\frac{\partial G_E}{\partial Q^{st}_{a_s c_t}}, \quad \hat{m}^t_{a_t} = \alpha\frac{1}{G_E}\frac{\partial G_E}{\partial m^t_{a_t}}, \tag{77}$$

from the saddle point conditions of $\hat{Q}^{st}_{a_s c_t}$, $\hat{m}^s_{a_t}$, $Q^{st}_{a_s c_t}$ and $m^s_{a_t}$, respectively, and

$$\frac{1}{G_E}\frac{\partial G_E}{\partial B^t_{a_t}} = 0, \tag{78}$$

from the saddle point condition of $B^t_{a_t}$.

In the limit where $n^1, \cdots n^T \to 0$, we have $G_E \to 1$ and $G_S \to 1$. Therefore, Eq. (77) specifically become

$$Q^{a_s c_t}_{a_s c_t} = \mathbb{E}_{\boldsymbol{w}}\left[w^s_{a_s} w^t_{c_t}\right] \tag{79}$$

$$m^t_{a_t} = \mathbb{E}_{\boldsymbol{w}}\left[w^t_{a_t}\right] \tag{80}$$

$$\hat{Q}^{st}_{a_s c_t} = 2\alpha\frac{\partial}{\partial Q^{st}_{a_s c_t}}\mathbb{E}_{\boldsymbol{u}, y, y^{\text{true}}}\exp\left[-\sum_{t=0}^T \gamma^t \sum_{a_t}\ell\left(v^{t-1}_1, v^t_{a_t}\right)\right] \tag{81}$$

$$= \Delta\alpha\mathbb{E}_{\boldsymbol{u}, y, y^{\text{true}}}\frac{\partial}{\partial u^s_{a_s} u^t_{c_t}}\exp\left[-\sum_{t=0}^T \gamma^t \sum_{a_t}\ell\left(v^{t-1}_1, v^t_{a_t}\right)\right] \tag{82}$$

$$\hat{m}^t_{a_t} = \alpha\frac{\partial}{\partial m^t_{a_t}}\mathbb{E}_{\boldsymbol{u}, y, y^{\text{true}}}\exp\left[-\sum_{t=0}^T \gamma^t \sum_{a_t}\ell\left(v^{t-1}_1, v^t_{a_t}\right)\right] \tag{83}$$

$$= \alpha\mathbb{E}_{\boldsymbol{u}, y, y^{\text{true}}}\frac{\partial}{\partial u^t_{a_t}}\exp\left[-\sum_{t=0}^T \gamma^t \sum_{a_t}\ell\left(v^{t-1}_1, v^t_{a_t}\right)\right], \tag{84}$$

where we assumed the random variables $\left\{w^t_{a_t}\right\}_{t, a^t}$ in Eqs. (79) and (81) follow the following distribution:

$$p(\boldsymbol{w}) \propto \exp\left[\sum_t\sum_{a_t}\frac{\gamma^t}{2}\lambda^t\left(w^t_{a_t}\right)^2 + \sum_t\sum_{a_t}\hat{m}^t_{a_t}w^t_{a_t} + \frac{1}{2}\sum_{st}\sum_{a_s c_t}\hat{Q}^{st}_{a_s c_t}w^s_{a_s}w^t_{c_t}\right], \tag{85}$$

and

$$v^t_{a_t} = \sigma\left((2y^{\text{true}} - 1)m^t_{a_t} + u^t_{a_t} + B^t_{a_t}\right) \quad (t \geq 0) \tag{86}$$

$$v^{-1}_1 = y. \tag{87}$$

## C.6 RS assamption

From Eqs. (61) and (74), the solutions of saddle point equations are related to the statistical properties of the replica variables, i.e.,

$$m^t_{a_t} = \frac{1}{N}\sum_i\mathbb{E}\left[w^t_{a_t, i}\right], \quad Q^{st}_{a_s c_t} = \frac{1}{N}\sum_i\mathbb{E}\left[w^s_{a_s, i}w^t_{c_t, i}\right] \tag{88}$$

in $n^1, \cdots, n^T \to 0$ limit, with the expectation taken over the probability distribution defined by Eq. (61). From the fact that the $p$-body correlation functions between replicas correspond to the $p$-th moments $\mathcal{F}_p$ in the original Markov process, we obtain

$$\mathcal{F}_{\boldsymbol{e}^{(t)}} = \mathbb{E}_{\mathcal{D}}\big[\hat{w}_i^t\big] = m_1^t \tag{89}$$

$$\mathcal{F}_{\boldsymbol{e}^{(s,t)}} = \mathbb{E}_{\mathcal{D}}\big[\hat{w}_i^s \hat{w}_i^t\big] = Q_{a_s c_t}^{st}, \tag{90}$$

where $\boldsymbol{e}^{(t)}$ and $\boldsymbol{e}^{(s,t)}$ are $T$-dimensional vectors defined as follows:

$$\boldsymbol{e}^{(t)} = (e_1, e_2, \ldots, e_T), \quad \text{where } e_i = \begin{cases} 1 & \text{if } i = t, \\ 0 & \text{otherwise}, \end{cases}$$

and

$$\boldsymbol{e}^{(s,t)} = (e_1, e_2, \ldots, e_T), \quad \text{where } e_i = \begin{cases} 1 & \text{if } i = s \text{ or } i = t, \\ 0, & \text{otherwise}, \end{cases}$$

and $(a_s, a_t) = (1, 2)$ if $s = t = T$, and $(a_s, a_t) = (1, 1)$ otherwise. Equation (90) holds for an arbitrary $i$ since the integrand of Eq. (69) can be written independently of $i$.

However, the replica parameters $(m_a^t, Q_a^{st}, \hat{m}_a^t$ and $\hat{Q}_{ab}^{st})$ are ill-defined in the limit as $n^1, \cdots, n^T \to 0$. To further advance our calculations and obtain well-defined quantities, we invoke the replica symmetry (RS) assumption, which posits a symmetry under permutation between different replicas, i.e.,

$$m_a^t = m^t \tag{91}$$

$$Q_{a_t c_t}^{tt} = Q^{tt} + \frac{\chi^{tt}}{\gamma^t} \delta_{a_t c_t} \tag{92}$$

$$Q_{a_s c_t}^{st} = Q^{st} + \frac{\chi^{st}}{\gamma^s} \delta_{a_s 1} \quad (s \neq t) \tag{93}$$

$$\hat{m}_{a_t}^t = \gamma^t \hat{m}^t \tag{94}$$

$$\hat{Q}_{a_t c_t}^{tt} = \left(\gamma^t\right)^2 \hat{\chi}^{tt} - \gamma^t \hat{Q}^{tt} \delta_{a_s c_t} \tag{95}$$

$$\hat{Q}_{a_s c_t}^{st} = \gamma^s \gamma^t \hat{\chi}^{st} - \gamma^t \hat{Q}^{st} \delta_{a_s 1} \quad (s \neq t) \tag{96}$$

$$B_{a_t}^t = b^t \tag{97}$$

where $\delta_{ab}$ is the Kronecker delta function. Using this parameterization, one can deduce

$$m^t = \mathbb{E}\big[\hat{w}_i^t\big] \tag{98}$$

$$Q^{st} = \mathbb{E}\big[\hat{w}_i^s \hat{w}_i^t\big], \tag{99}$$

for an arbitrary $i$. Higher-order moments can be immediately determined to be zero from Eq. (69), given that the distribution in Eq. (61) indicates that $\boldsymbol{w}^1, \cdots, \boldsymbol{w}^T$ follow a Gaussian distribution. Although the RS assumption is not mathematically rigorous in general, it has been empirically validated in many practical scenarios, particularly in convex optimization problems. To date, there are no known examples where the RS assumption leads to incorrect predictions in convex settings.

## C.7 Saddle point equations: order parameters

Our next step is to derive the saddle-point equations for the order parameters $(m^t, Q^{st}$ and $\chi^{st})$ under the RS assamption from the ones for replica parameters $(\hat{m}_a^t, \hat{Q}_{ab}^{st}$ and $\hat{\chi}_{ab}^{st})$. First, substitution of the

RS assumption in eq. (85) yields

$$p(\boldsymbol{w}) \propto \exp\left[-\sum_t \gamma_t \frac{\hat{Q}^{tt}+\lambda^t}{2}\sum_{a_t}(w_{a_t}^t)^2 + \sum_t \gamma^t \hat{m}^t \sum_{a_t} w_{a_t}^t + \sum_{(s<t)} \gamma^t \hat{Q}^{st} w_1^{a_s}\sum_{a_t} w_{a_t}^t\right.$$

$$\left.+\frac{1}{2}\sum_{st}\hat{\chi}^{st}\left(\gamma^s \sum_{a_s} w_{a_s}^s\right)\left(\gamma^t \sum_{a_t} w_{a_t}^t\right)\right] \tag{100}$$

$$=\int D\hat{\boldsymbol{\xi}}\exp\left[-\sum_t \gamma_t \frac{\hat{Q}^{tt}+\lambda^t}{2}\sum_{a_t}(w_a^t)^2 + \sum_t \gamma^t \hat{m}^t \sum_{a_t} w_a^t + \sum_{(s<t)}\gamma^t\hat{Q}^{st}w_1^s\sum_a w_{a_t}^t\right.$$

$$\left.+\sum_{st}\gamma_s\sum_{a_s}w_{a_s}^s\sqrt{\hat{\chi}}^{st}\xi^t\right] \tag{101}$$

$$=\int D\hat{\xi}^0 \exp\left[-\gamma^1 \sum_{a_0}\left(\frac{\hat{Q}^{00}+\lambda^0}{2}(w_{a_0}^0)^2 - \left(\hat{m}^1 + \sum_s \sqrt{\hat{\chi}}^{0s}\xi^s\right)w_{a_0}^0\right)\right]$$

$$\times \prod_{t=1}^T \int D\hat{\xi}^t \exp\left[-\gamma^t \sum_{a_t}\left(\frac{\hat{Q}^{tt}+\lambda^t}{2}(w_{a_t}^t)^2 - \left(\hat{m}^t + \sum_s \sqrt{\hat{\chi}}^{ts}\xi^s - \sum_{s=0}^{t-1}\hat{Q}^{st}w_1^s\right)w_{a_t}^t\right)\right], \tag{102}$$

where $\sqrt{\hat{\chi}}^{st}$ is the $s,t$ element of the cholesky decomposition of the matrix $\hat{\chi}$, and $D\hat{\xi}^t$ is the normal Gaussian measure.

Now we define following notations:

$$f^t(w_a^t; w_1^0, \ldots, w_1^{t-1}) = \begin{cases} \frac{\hat{Q}^{00}+\lambda^0}{2}(w_a^0)^2 - \left(\hat{m}^0 + \sum_s \sqrt{\hat{\chi}}^{0s}\xi^s\right)w_a^0 & (t=0) \\ \frac{\hat{Q}^{tt}+\lambda^t}{2}(w_a^t)^2 - \left(\hat{m}^t + \sum_s \sqrt{\hat{\chi}}^{ts}\xi^s - \sum_{s=1}^{t-1}\hat{Q}^{st}w_1^s\right)w_a^t & (t\geq 1) \end{cases} \tag{103}$$

$$w_*^t(w_1^0, \cdots, w_1^{t-1}) = \underset{w^t}{\mathrm{argmin}}\, f^t(w^t; w_1^0, \cdots, w_1^{t-1}) = \begin{cases} \frac{\hat{m}^0 + \sum_s \sqrt{\hat{\chi}}^{0s}\xi^s}{\lambda^1 + \hat{Q}^{00}} & (t=0) \\ \frac{\hat{m}^t + \sum_s \sqrt{\hat{\chi}}^{ts}\xi^s - \sum_{s=0}^{t-1}\hat{Q}^{st}w_1^s}{\lambda^t + \hat{Q}^{tt}} & (t\geq 1) \end{cases} \tag{104}$$

$$w_*^t = w_*^t(w_*^0, w_*^1(w_*^0), \cdots, w_*^{t-1}(w_*^0, \cdots, w_*^{t-2})) \tag{105}$$

$$w_*^t(w^0, \cdots, w_1^s) = w_*^t(w_1^0, \cdots, w_1^s, w_*^{s+1}(w^0, \cdots, w^s)). \tag{106}$$

Then, the saddle-point equations of order parameters are simplified as (in the $n^1, \cdots, n^T \to 0, \gamma^1, \cdots, \gamma^T \to 0$ limit)

$$m^t = \mathbb{E}[w_a^t] \tag{107}$$

$$=\int D\hat{\boldsymbol{\xi}} \prod_{s=0}^{t-1}\int d\boldsymbol{w}^s \exp(-\gamma^s f^s)\frac{\int dw^t\, w^t \exp(-\gamma^t f^t)}{\int dw^t \exp(-\gamma^t f^t)} \tag{108}$$

$$=\mathbb{E}_{\hat{\boldsymbol{\xi}}}[w_*^t] \tag{109}$$

$$Q^{st} = \mathbb{E}[w_{a_s}^s w_{c_t}^t] \tag{110}$$

$$=\int D\hat{\boldsymbol{\xi}} \prod_{l=0}^{s-1}\int d\boldsymbol{w}^l \exp(-\gamma^l f^l)\frac{\int dw^s\, w^s \exp(-\gamma^s f^s)}{\int dw^s \exp(-\gamma^s f^s)}$$

$$\times \prod_{l=s+1}^{t-1}\int d\boldsymbol{w}^l \exp(-\gamma^l f^l)\frac{\int dw^t\, w^t \exp(-\gamma^t f^t)}{\int dw^t \exp(-\gamma^t f^t)} \tag{111}$$

$$=\mathbb{E}_{\hat{\boldsymbol{\xi}}}[w_*^s w_*^t] \quad (s<t) \tag{112}$$

$$Q^{tt} = \mathbb{E}\left[w_{a_t}^t w_{c_t}^t\right] \tag{113}$$

$$= \int \mathrm{D}\hat{\boldsymbol{\xi}} \prod_{s=0}^{t-1} \int \mathrm{d}\boldsymbol{w}^s \, \exp(-\gamma^s f^s) \left( \frac{\int \mathrm{d}w^t \, w^t \exp(-\gamma^t f^t)}{\int \mathrm{d}w^t \, \exp(-\gamma^t f^t)} \right)^2 \tag{114}$$

$$= \mathbb{E}_{\hat{\boldsymbol{\xi}}}\left[(w_*^t)^2\right] \tag{115}$$

$$\chi^{st} = \gamma^s \mathbb{E}\left[w_{a_s}^s w_{c_t}^t - w_1^s w_{c_t}^t\right] \tag{116}$$

$$= \gamma^s \int \mathrm{D}\hat{\boldsymbol{\xi}} \prod_{l=0}^{s-1} \int \mathrm{d}\boldsymbol{w}^l \, \exp(-\gamma^l f^l) \left[ \frac{\int \mathrm{d}w_1^s \, w_1^s \exp(-\gamma^s f^s(w_1^s))}{\int \mathrm{d}w^s \, \exp(-\gamma^s f^s(w^s))} w_*^t(w_1^1, \cdots, w_1^s) \right.$$
$$\left. - \frac{\int \mathrm{d}w^s \, w^s \exp(-\gamma^s f^s(w^s)) \int \mathrm{d}w_1^s \, w_*^t(w_1^1, \cdots, w_1^s)}{\left(\int \mathrm{d}w^s \, \exp(-\gamma^s f^s)\right)^2} \right] \tag{117}$$

$$= \int \mathrm{D}\hat{\boldsymbol{\xi}} \prod_{l=0}^{s-1} \int \mathrm{d}\boldsymbol{w}^l \, \exp(-\gamma^l f^l) \frac{\mathrm{d}}{\mathrm{d}\hat{m}^s} \frac{\int \mathrm{d}w_1^s \, \exp(-\gamma^s f^s(w_1^s)) w_*^t(w_1^1, \cdots, w_1^s)}{\int \mathrm{d}w^s \, \exp(-\gamma^s f^s)} \tag{118}$$

$$= \mathbb{E}_{\hat{\boldsymbol{\xi}}}\left[ \frac{\mathrm{d}w_*^t}{\mathrm{d}\hat{m}^s} \right] \quad (s < t) \tag{119}$$

$$\chi^{tt} = \gamma^t \mathbb{E}\left[ \left(w_{a_t}^t\right)^2 - w_{a_t}^t w_{c_t}^t \right] \tag{120}$$

$$= \gamma^t \int \mathrm{D}\hat{\boldsymbol{\xi}} \prod_{s=0}^{t-1} \int \mathrm{d}\boldsymbol{w}^s \, \exp(-\gamma^s f^s) \left[ \frac{\int \mathrm{d}w^t \, (w^t)^2 \exp(-\gamma^t f^t)}{\int \mathrm{d}w^t \, \exp(-\gamma^t f^t)} - \left( \frac{\int \mathrm{d}w^t \, w^t \exp(-\gamma^t f^t)}{\int \mathrm{d}w^t \, \exp(-\gamma^t f^t)} \right)^2 \right] \tag{121}$$

$$= \int \mathrm{D}\hat{\boldsymbol{\xi}} \prod_{s=0}^{t-1} \int \mathrm{d}\boldsymbol{w}^s \, \exp(-\gamma^s f^s) \frac{\partial}{\partial \hat{m}^t} \frac{\int \mathrm{d}w^t \, w^t \exp(-\gamma^t f^t)}{\int \mathrm{d}w^t \, \exp(-\gamma^t f^t)} \tag{122}$$

$$= \mathbb{E}_{\hat{\boldsymbol{\xi}}}\left[ \frac{\partial w_*^t}{\partial \hat{m}^t} \right]. \tag{123}$$

By introducing the helper variable $R^{st} = \mathbb{E}_{\hat{\boldsymbol{\xi}}}\left[ \sqrt{\bar{\chi}}^{ts} \hat{\xi}^s \hat{w}_*^t \right]$ for simplicity, the explicit calculation of these equations yields

$$R^{st} = \mathbb{E}\left[ \sqrt{\hat{\chi}}^{ts} \hat{\xi}^s \hat{w}_*^t \right] = \begin{cases} \frac{1}{\hat{Q}^{00}+\lambda^0} \hat{\chi}^{s0} & (t = 0) \\ \frac{1}{\hat{Q}^{tt}+\lambda^t} \left( \hat{\chi}^{st} - \sum_{l=0}^{t-1} \hat{Q}^{lt} R^{sl} \right) & (t \geq 1) \end{cases} \tag{124}$$

$$Q^{st} = \mathbb{E}\left[ \hat{w}_*^s \hat{w}_*^t \right] = \begin{cases} \frac{1}{\hat{Q}^{00}+\lambda^0} \left( \hat{m}^0 m^t + R^{0t} \right) & (s = 0) \\ \frac{1}{\hat{Q}^{ss}+\lambda^s} \left( \hat{m}^s m^t + R^{st} - \sum_{l=0}^{s-1} \hat{Q}^{ls} Q^{lt} \right) & (s \geq 1) \end{cases} \tag{125}$$

$$m^t = \mathbb{E}\left[ \hat{w}_*^t \right] = \begin{cases} \frac{1}{\hat{Q}^{00}+\lambda^0} \hat{m}^0 & (t = 0) \\ \frac{1}{\hat{Q}^{tt}+\lambda^t} \hat{m}^t \left( \hat{m}^t - \sum_{s=0}^{t-1} \hat{Q}^{st} m^s \right) & (t \geq 1) \end{cases} \tag{126}$$

$$\chi^{st} = \mathbb{E}\left[ \frac{\mathrm{d}w_*^t}{\mathrm{d}\hat{m}^s} \right] = \begin{cases} \frac{1}{\hat{Q}^{tt}+\lambda^t} & (s = t) \\ -\frac{1}{\hat{Q}^{tt}+\lambda^t} \sum_{l=0}^{s} \hat{Q}^{lt} \chi^{sl} & (s < t) \end{cases} . \tag{127}$$

## C.8 Saddle point equations: conjugate parameters

Similar to the previous section, we now derive the saddle-point equations for the conjugate parameters $(m^t, q^{st}, \chi^{st})$ based on the RS assumption.

The covariance matrix of the Gaussian variables $\boldsymbol{u}$ (Eq. (63)) is rewritten as

$$\mathbb{E}_{\mathcal{D}}\left[u_{a_t}^t u_{c_t}^t\right] = \Delta\left(Q^{tt} + \frac{\chi^{tt}}{\gamma^t}\delta_{a_t c_t}\right) \tag{128}$$

$$\mathbb{E}_{\mathcal{D}}\left[u_{a_s}^s u_{c_t}^t\right] = \Delta\left(Q^{st} + \frac{\chi^{st}}{\gamma^s}\delta_{a_s 1}\right) \quad (s < t). \tag{129}$$

Under these conditions, we can introduce the random variable $\tilde{\boldsymbol{u}}$ with an equivalent distribution as follows:

$$\tilde{u}_a^t = \sum_{r=0}^t A_{tr}\xi_0^r + \sum_{r=0}^t \frac{\chi^{rt}}{\chi^{rr}}z_1^r, \tag{130}$$

where $A_{st}$ are the cholesky decomposition of the covariance matrix of $\boldsymbol{u}$, i.e., $\sum_r A_{sr}A_{tr} = \Delta Q^{st}$, $z_a^t = \sqrt{\Delta\chi^{tt}/\gamma^t}\xi_a^t$, and $\xi_0^t, \xi_a^t \sim \mathcal{N}(0,1)$ are independent standard normal random variables.

Following the same procedure as the previous section, taking the limit of $\gamma^t \to \infty$ in order, the expectation calculation is transformed into the solution of the optimization problem, and finally the following relationship is obtained:

$$\hat{Q}^{st} = -\frac{\alpha}{\chi^{tt}}\mathbb{E}_{y,y^{\text{true}},\boldsymbol{\xi}}\left[\frac{\mathrm{d}z_*^t}{\mathrm{d}h^s}\right] \tag{131}$$

$$\hat{m}^t = \frac{\alpha}{\Delta\chi^{tt}}\mathbb{E}_{y,y^{\text{true}},\boldsymbol{\xi}}\left[(2y-1)z_*^t\right] \tag{132}$$

$$\hat{\chi}^{st} = \frac{\alpha}{\Delta\chi^{ss}\chi^{tt}}\mathbb{E}_{y,y^{\text{true}},\boldsymbol{\xi}}\left[z_*^s z_*^t\right] \tag{133}$$

$$\mathbb{E}_{y,y^{\text{true}},\boldsymbol{\xi}}\left[z_*^t\right] = 0, \tag{134}$$

where $z_*^t$ is the solution of the optimization problem as follows:

$$z_*^0 = \underset{z^0}{\operatorname{argmin}}\left[\frac{(z^0)^2}{2\Delta\chi^{00}} + \ell\left(y^0, h^0 + z^0\right)\right] \tag{135}$$

$$h^0 = A_{00}\xi_1^0 + (2y^{\text{true}} - 1)m^0 + b^0 \tag{136}$$

$$z_*^t = \underset{z^t}{\operatorname{argmin}}\left[\frac{(z^t)^2}{2\Delta\chi^{tt}} + \ell\left(\sigma\left(\beta^t\left(h^{t-1} + z_*^{t-1}\right)\right), \sigma\left(h^t + z^t\right)\right)\right] \quad (1 \leq t \leq T) \tag{137}$$

$$h^t = \sum_{s=0}^t A_{st}\xi_0^s + \sum_{s=0}^{t-1} B_{st}z_*^s + (2y^{\text{true}} - 1)m^t + b^t \quad (1 \leq t \leq T). \tag{138}$$

## C.9 Derivation of Result C.1

To summarize the results obtained so far, we have derived the first- and second-order statistics of the estimator $\hat{w}_i^t$, which are determined by the constants $m^t$ and $Q^{st}$ (Eqs. (98) and (99)). Furthermore, we have shown that these constants can be computed by solving the saddle-point equations defined in Eqs. (124)–(127) and Eqs. (131)–(134). As a representation of the distribution of $\hat{w}_i^t$ that satisfies all the conditions for the integer moments, we can express it as

$$\hat{w}_i^0 \stackrel{\mathrm{d}}{=} \frac{1}{\hat{Q}^{00} + \lambda^0}\left(\hat{m}^0 + \hat{\xi}^0\right) \tag{139}$$

$$\hat{w}_i^t \stackrel{\mathrm{d}}{=} \frac{1}{\hat{Q}^{tt} + \lambda^t}\left(\hat{m}^t + \hat{\xi}^t - \sum_{s=0}^{t-1}\hat{Q}^{st}\hat{w}^s\right) \quad (t \geq 1) \tag{140}$$

and this representation is unique up to equivalent forms. Furthermore, by proceeding with similar calculations while taking into account the correlation with the data, Eqs. (135) and (137) yield

$$\frac{\hat{\boldsymbol{w}}^t \cdot \boldsymbol{x}_\mu}{\sqrt{N}} + \hat{B}^t \stackrel{\mathrm{d}}{=} h^t + z_*^t \tag{141}$$

for the pre-activation distribution.

## C.10 Remarks on Rigorous Proofs

Beyond the replica method, a fully rigorous proof in our setting remains open. The main difficulty is the temporal correlation across multiple SD stages. AMP/state-evolution techniques Donoho et al. [2009], Liu and Ma [2024] have been applied successfully to handle such correlations, but basically only for first-order iterative algorithms, and thus do not directly extend to multi-stage SD. CGMT-based approaches Thrampoulidis et al. [2015] could in principle be adapted, but they are typically used in on-line settings where new data are introduced at each stage. Developing tools that combine these techniques to rigorously capture the multi-stage dynamics would be an interesting direction for future work.

## C.11 Remarks on Numerical Calculations

**Iterative Method** In numerical calculations of the saddle point equations Eqs. (124)-(127) and Eqs. (131)-(134) in replica method, the order parameters are usually obtained through iterative method. The procedure starts from an initial guess of the order parameters. These values are substituted into the left-hand side of the saddle-point equations, and the resulting right-hand side gives an updated estimate of the parameters. This process is repeated until the values converge within a chosen tolerance. In this way, the order parameters are determined as the fixed point of the equations.

**Discrete Expectation** When the theoretical formulation includes averages over discrete random variables such as the class label and the label-flip noise, each possible configuration is evaluated separately. In the present model, the class label $y$ takes values $0$ or $1$, and the label may be either flipped or unflipped. Hence, there are four possible combinations: $(y, y_{\text{true}}) = (0,0), (0,1), (1,0), (1,1)$. For a generic quantity $f(y, y_{\text{true}})$, the expectation appearing in the saddle-point equations is computed as

$$\mathbb{E}_{y, y_{\text{true}}}[f(y, y_{\text{true}})] = \rho(1-\theta)f(0,0) + (1-\rho)\theta f(0,1) + \rho(1-\theta)f(1,0) + (1-\rho)(1-\theta)f(1,1).$$
(142)

In the numerical implementation, the value of $f(y, y_{\text{true}})$ is computed and maintained for all four cases.

**Sequential Update in Time** The calculation also proceeds sequentially with respect to the time index $t$. The result at step $t$ depends only on the quantities determined at earlier steps from $0 \to t-1$. Therefore, the parameters can be updated step by step, starting from $t = 0$. This recursive approach makes it straightforward to implement the computation and to monitor convergence at each stage.

**Evaluation of Total Derivatives in Eq.** (131) We describe the implementation of evaluating the total derivatives appearing on the left-hand side of Eq. (131) during the numerical calculation of the saddle point equations.

For notational simplicity, we define the partial derivatives of the loss function $\ell$ as

$$\ell_t' = \frac{\partial \ell(\sigma(h^{t-1} + z_*^{t-1}), \sigma(h^t + z^t))}{\partial h^t}, \quad \bar{\ell}_t = \frac{\partial \ell(\sigma(h^{t-1} + z_*^{t-1}), \sigma(h^t + z^t))}{\partial h^{t-1}}.$$
(143)

Introducing the function

$$F^t(z^t, z_*^{t-1}, h^t, h^{t-1}) = \frac{z^t}{\Delta \chi^{tt}} + \ell_t',$$
(144)

we have the condition

$$F^t(z_*^t, z_*^{t-1}, h^t, h^{t-1}) = 0.$$
(145)

By employing the implicit function theorem, we obtain

$$\frac{\partial F^t}{\partial z_*^t}\frac{\mathrm{d}z_*^t}{\mathrm{d}h^s} + \frac{\partial F^t}{\partial h^{t-1}}\frac{\mathrm{d}h^{t-1}}{\mathrm{d}h^s} + \frac{\partial F^t}{\partial h^t}\frac{\mathrm{d}h^t}{\mathrm{d}h^s} + \frac{\partial F^t}{\partial z_*^{t-1}}\frac{\mathrm{d}z_*^{t-1}}{\mathrm{d}h^s} = 0.$$
(146)

since $z_*^t$ represents the solution of the optimization problem given by Eq. (137). Solving for $\frac{\mathrm{d}z_*^t}{\mathrm{d}h^s}$, we obtain

$$\frac{\mathrm{d}z_*^t}{\mathrm{d}h^s} = a^t \frac{\mathrm{d}h^t}{\mathrm{d}h^s} + b^t \frac{\mathrm{d}h^{t-1}}{\mathrm{d}h^s} + c^t \frac{\mathrm{d}z_*^{t-1}}{\mathrm{d}h^s},$$
(147)

where the coefficients $a^t, b^t, c^t$ are given by

$$a^t = -\frac{\frac{\partial F^t}{\partial h^t}}{\frac{\partial F^t}{\partial z_*^t}} = -\frac{\ell''}{\ell'' + 1/(\Delta\chi^{tt})}, \tag{148}$$

$$b^t = -\frac{\frac{\partial F^t}{\partial h^{t-1}}}{\frac{\partial F^t}{\partial z_*^t}} = -\frac{\bar{\ell}'}{\ell'' + 1/(\Delta\chi^{tt})}, \tag{149}$$

$$c^t = -\frac{\frac{\partial F^t}{\partial z_*^{t-1}}}{\frac{\partial F^t}{\partial z_*^t}} = -\frac{\bar{\ell}'}{\ell'' + 1/(\Delta\chi^{tt})} = b^t. \tag{150}$$

Furthermore, using the following relations,

$$\frac{\mathrm{d}h^t}{\mathrm{d}h^s} = \sum_{r<t} \frac{\chi^{rt}}{\chi^{rr}} \frac{\mathrm{d}z_*^r}{\mathrm{d}h^s}, \tag{151}$$

$$\frac{\mathrm{d}z_*^t}{\mathrm{d}h^s} = \sum_{r<t-1} \frac{\chi^{r,t-1}}{\chi^{r,t-1}} \frac{\mathrm{d}z_*^r}{\mathrm{d}h^s}, \tag{152}$$

one can iteratively compute $\frac{\mathrm{d}z_*^t}{\mathrm{d}h^s}$ for a fixed $t$, as illustrated in Algorithm 1, which presents a single iteration step of the update procedure. The full derivative is obtained by repeatedly applying this update until convergence.

---

**Algorithm 1: UpdateQhatColumn**$(t)$: Self-consistent update of column $t$ (past columns are fixed)

---

**Input:** Results up to time $t - 1$:

$$\left\{ z_*^{1:t-1}, \ h^{1:t-1}, \ G[s,r] = \tfrac{dz_*^r}{dh^s}, \ H[s,r] = \tfrac{dh^r}{dh^s} \ (s \le r \le t-1) \right\} \tag{153}$$

**Output:** Results at time $t$:

$$\left\{ G[s,t] = \tfrac{dz_*^t}{dh^s}, \ H[s,t] = \tfrac{dh^t}{dh^s} \ (s \le t), \ \hat{Q}[s,t] \ (s \le t) \right\} \tag{154}$$

1 **Step 1: Calculate** $a^t, b^t, c^t$
2

$$a^t \leftarrow -\frac{\ell''_{t,t}}{\ell''_{t,t} + 1/(\Delta\chi^{t,t})} \tag{155}$$

$$b^t \leftarrow -\frac{\ell''_{t,t-1}}{\ell''_{t,t} + 1/(\Delta\chi^{t,t})} \tag{156}$$

$$c^t \leftarrow b^t \tag{157}$$

3 **Step 2: Calculate** $H[t,s]$
4 **for** $s \leftarrow 1$ **to** $t - 1$ **do**
5 $\quad\lfloor \ H[s,t] \leftarrow \sum_{r<t} \frac{\chi^{rt}}{\chi^{rr}} G[s,r]$
6 $H[t,t] \leftarrow 1$

7 **Step 3: Calculate** $G[t,s]$
8 **for** $s \leftarrow 1$ **to** $t - 1$ **do**
9 $\quad\lfloor \ G[s,t] \leftarrow a^t H[s,t] + b^t H[s,t-1] + c^t G[s,t-1]$
10 $G[t,t] \leftarrow a^t$

11 **Step 5: Update** $\hat{Q}[t,s]$;
12 **for** $s \leftarrow 1$ **to** $t$ **do**
13 $\quad\lfloor \ \hat{Q}_{\text{new}}[s,t] \leftarrow -\frac{\alpha}{\chi^{tt}} \mathbb{E}[G[s,t]]$

---

## D   Theoretical and Experimental Validation

In this appendix, we present evidence demonstrating the strong agreement between theoretical predictions derived from the replica method and numerical experiments for the linear $t$-SD model. Figure 7 compares the generalization error, weight distribution, and pre-activation distribution obtained from replica method with those from numerical experiments, revealing remarkable consistency between the two approaches.

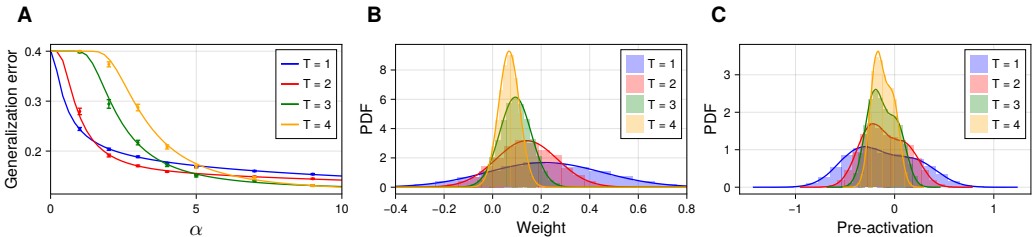

Figure 7: Comparison of theoretical predictions derived from replica method and numerical experiments for the linear $t$-SD model statistics. (A) Generalization error derived by the replica method (solid lines) and numerical simulations (dots with error bars). (B) Distributions of optimal weights derived by the replica method (solid lines) and their empirical distributions obtained from a single experiment (histograms). (C) Pre-activation distributions predicted by theory (solid lines) and empirically observed from a single experiment (histograms). Parameters for (A-C): $\rho = 0.4, \Delta = 0.6, \theta = 0.2, (\lambda_1, \lambda_2, \lambda_3, \lambda_4) = (1.5, 0.5, 2.0, 1.0), (\beta_1, \beta_2, \beta_3) = (0.8, 1.2, 1.0)$; (B, C) $\alpha = 3.0$. Numerical experiments: (A) $N = 10^3$ ( Error bars represent the standard error of the mean over 20 trials per point.); (B, C) $N = 10^4$.

## E   Optimization of the hyper parameters

The above results describe the statistical properties of the estimators $\{\hat{\boldsymbol{w}}^t, \hat{B}^t\}_{t \geq 0}$ and the generalization error for a fixed hyperparameters $\{\lambda^t\}_{t \geq 0}, \{\beta^t\}_{t \geq 1}$. In order to find the optimal hyper parameters, we used the Nelder-Mead (NM) method Dixit and Rackauckas [2023], which is a versatile black-box optimization algorithm. At each optimization stage in NM, we numerically solve the set of equations in Result C.1, which can be efficiently solved using a simple fixed point iteration to evaluate the generalization error (9).

## F   Exact results for the linear $t$-SD model

In this appendix, we present several simplified analyses of the linear $t$-SD model and conclude by proving a generalized version of Result 2.

### F.1   Integrated saddle point equations in the linear $t$-SD model

In the case of the mean squared error loss and the linear activation function, the saddle point equations for the conjugate variables and $b^t$ is integrable. For some algebraic manipulations, we have the following equations:

$$z_*^t = \frac{\Delta\chi^{tt}}{2 + \Delta\chi^{tt}}\left(2y^{t-1} - h^t - 1\right) \tag{158}$$

$$y^t = \frac{1}{2}\left(\beta\left(h^t + z_*^t\right) + 1\right) \quad (t \geq 1) \tag{159}$$

$$\hat{Q}^{tt} = \frac{\Delta\alpha}{2 + \Delta\chi^{tt}} \tag{160}$$

$$\hat{Q}^{st} = \frac{\Delta\alpha}{2 + \Delta\chi^{tt}} \left[ -\beta^{t-1}\left(\delta^{t-1,s} - \frac{1}{\alpha}\sum_{l=s}^{t-1}\chi^{l,t-1}\hat{Q}^{sl}\right) - \frac{1}{\alpha}\sum_{l=s}^{t-1}\chi^{lt}\hat{Q}^{sl} \right] \quad (s < t) \tag{161}$$

$$\tag{162}$$

$$\hat{m}^0 = \frac{2\alpha\rho}{2 + \Delta\chi^{00}}\left(2(1-\theta) - (m^0 + b^0) - 1\right) \tag{163}$$

$$\hat{m}^t = \frac{2\alpha}{2 + \Delta\chi^{tt}}\left[\frac{\Delta}{2\alpha}\left(\sum_{s=0}^{t-1}(\beta^{t-1}\chi^{s,t-1} - \chi^{st})\hat{m}^s\right) + \beta^{t-1}\rho(m^{t-1} + b^{t-1}) - \rho(m^t + b^t)\right] \quad (1 \le t \le T) \tag{164}$$

$$\hat{\chi}^{0t} = \frac{\Delta}{2 + \Delta\chi^{00}}\left[\frac{2\alpha}{\Delta\chi^{tt}}\hat{r}^t + \left(\sum_{l=0}^{t}Q^{0l}\hat{Q}^{lt} - m^0\hat{m}^t\right)\right], \tag{165}$$

$$\hat{\chi}^{st} = \frac{\Delta}{2 + \Delta\chi^{ss}}\left[-\beta^{s-1}\left(\sum_{l=0}^{t}Q^{\min(s-1,l),\max(s-1,l)}\hat{Q}^{lt} - \sum_{l=0}^{s-1}\chi^{l,s-1}\hat{\chi}^{lt} - m^{s-1}\hat{m}^t\right) \right. \tag{166}$$

$$\left. + \left(\sum_{l=0}^{t}Q^{sl}\hat{Q}^{lt} - \sum_{l=0}^{s-1}\chi^{ls}\hat{\chi}^{lt} - m^s\hat{m}^t\right)\right]. \tag{167}$$

$$\hat{r}^0 = \frac{\Delta\chi^{00}}{2 + \Delta\chi^{00}}\left[(\rho + \theta - 2\rho\theta) - \{\rho(1-\theta)(m^0 + b^0) + (1-\rho)\theta(-m^0 + b^0)\}\right] \tag{168}$$

$$\hat{r}^t = \frac{\Delta\chi^{tt}}{2 + \Delta\chi^{tt}}\left[\sum_{s=0}^{t-1}\frac{1}{\chi^{ss}}\left(\beta^{t-1}\chi^{s,t-1} - \chi^{st}\right)\hat{r}^s \right. \tag{169}$$

$$+ \beta^{t-1}\{\rho(1-\theta)(m^{t-1} + b^{t-1}) + (1-\rho)\theta(-m^{t-1} + b^{t-1})\} \tag{170}$$

$$\left. - \{\rho(1-\theta)(m^t + b^t) + (1-\rho)\theta(-m^t + b^t)\}\right] \quad (1 \le t \le T) \tag{171}$$

$$b^0 = 2(\rho + \theta - 2\rho\theta) - (2\rho - 1)m^0 - 1 \tag{172}$$

$$b^t = \beta^{t-1}\left((2\rho - 1)m^{t-1} + b^{t-1}\right) - (2\rho - 1)m^t \quad (1 \le t \le T), \tag{173}$$

where we introduced the auxiliary variable $\hat{r}^t = \mathbb{E}_{\mathcal{D}}\left[y^0 z_*^t\right]$ for simplicity.

### F.2 Case of $\rho = 1/2$ and $\lambda^0, \lambda^1 \to \infty$

One can solve the saddle point equations explicitly in some specific cases. For example, in the case of $\rho = 1/2$ and $\lambda^0, \lambda^1 \to \infty$, the explicit form of the genelization error is given by following proposition.

**Proposition F.1.** *In the linear t-SD model with $\rho = 1/2$ and $\lambda^0, \lambda^1 \to \infty$, the generalization errors at $t = 0$ and $t = 1$ are given by*

$$\mathcal{E}^0 = H\left(\frac{\sqrt{\alpha}(1 - 2\theta)}{\sqrt{\Delta\left(\Delta + \alpha(1 - 2\theta)^2\right)}}\right), \tag{174}$$

$$\mathcal{E}^1 = H\left(\frac{\alpha(\Delta + \alpha + \Delta\alpha)(1 - 2\theta)}{\sqrt{\Delta\left[(\alpha^2 + 3\alpha + 1)\Delta^3\alpha + \alpha^2\left(\Delta^2(\alpha^2 + 5\alpha + 3) + \Delta(2\alpha + 3)\alpha + \alpha^2\right)(1 - 2\theta)^2\right]}}\right). \tag{175}$$

*In particular, $\mathcal{E}^{*0} = \mathcal{E}^0$ and $\mathcal{E}^{*1} \le \mathcal{E}^1$.*

**F.3  Case of $\rho = 1/2$, $\lambda^0, \cdots, \lambda^T \to \infty$ and $T \to \infty$**

Another specific solvable case is $\lambda^0, \cdots, \lambda^T \to \infty$ and $T \to \infty$. For ease of reference, we first restate the generalized version of Result 2 as Result 3 below. We then proceed to its proof.

**Result 3.** *(The generalization error at $t \to \infty$) For an arbitrary choice of the set of the temperature parameters $\{\beta^t\}_{t \geq 0}$, the generalization error of the linear t-SD model with $\rho = 0.5$, $\lambda^0, \cdots, \lambda^t \to \infty$ and $t \to \infty$ is given by*

$$\lim_{t \to \infty} \mathcal{E}^t = \begin{cases} 0.5 & (\alpha < \Delta^2) \\ H\left(\sqrt{\frac{\alpha - \Delta^2}{\Delta(\alpha + \Delta)}}\right) & (\alpha \geq \Delta^2). \end{cases} \tag{176}$$

Under these conditions, equations (172) and (173) yield $b^0, \cdots, b^T = 0$. For simplicity, we set $\lambda^0 = \cdots = \lambda^T = \lambda$, $\epsilon = 1/\lambda$, and $\gamma^1 = \cdots = \gamma^T = \gamma$ without loss of generality. The scaling of each parameter with respect to $\epsilon$ and $\gamma$ is

$$m^t = \mathcal{O}(\epsilon^t)\mathcal{O}(\gamma^{t-1}) \qquad \hat{m}^t = \mathcal{O}(\epsilon^{t-1})\mathcal{O}(\gamma^{t-1}) \tag{177}$$

$$Q^{st} = \mathcal{O}(\epsilon^{s+t})\mathcal{O}(\gamma^{s+t-2}) \qquad \hat{Q}^{st} = \mathcal{O}(\epsilon^{t-s-1})\mathcal{O}(\gamma^{t-s}) \ (s < t), \ \hat{Q}^{tt} = \mathcal{O}(1)\mathcal{O}(1) \tag{178}$$

$$\chi^{st} = \mathcal{O}(\epsilon^{t-s+1})\mathcal{O}(\gamma^{t-s}) \qquad \hat{\chi}^{st} = \mathcal{O}(\epsilon^{t+s-2})\mathcal{O}(\gamma^{t+s-2}) \tag{179}$$

$$R^{st} = \mathcal{O}(\epsilon^{t+s-1})\mathcal{O}(\gamma^{t+s-2}). \tag{180}$$

Based on this scaling, we rescale each variable as

$$m^t \to \epsilon^t \gamma^{t-1} m^t \qquad \hat{m}^t \to \epsilon^{t-1} \gamma^{t-1} \hat{m}^t \tag{181}$$

$$Q^{st} \to \epsilon^{s+t} \gamma^{s+t-2} Q^{st} \qquad \hat{Q}^{st} \to \epsilon^{t-s-1} \gamma^{t-s} \hat{Q}^{st} \ (s < t), \ \hat{Q}^{tt} \to \hat{Q}^{tt} \tag{182}$$

$$\chi^{st} \to \epsilon^{t-s+1} \gamma^{t-s} \chi^{st} \qquad \hat{\chi}^{st} \to \epsilon^{t+s-2} \gamma^{t+s-2} \hat{\chi}^{st} \tag{183}$$

$$R^{st} \to \epsilon^{t+s-1} \gamma^{t+s-2} R^{st}. \tag{184}$$

Taking the limit as $\epsilon \to 0$, we obtain the following simplified recurrence relations:

$$\chi^{st} = \left(\frac{\Delta\alpha}{2}\right)^{t-s} \tag{185}$$

$$\hat{Q}^{tt} = \frac{\Delta\alpha}{2}, \quad \hat{Q}^{t-1,t} = -\frac{\Delta\alpha}{2} \tag{186}$$

$$\hat{Q}^{st} = -\left(\frac{\Delta(1+\alpha)}{2}\right)^{t-s-2} \frac{\Delta^2\alpha}{4} \quad (t \geq s \geq 2) \tag{187}$$

$$\hat{m}^0 = m^0 = \frac{\alpha}{2}(1 - 2\theta) \tag{188}$$

$$\hat{m}^1 = \frac{1}{2}(\Delta + \alpha)m^0, \quad m^1 = \frac{1}{2}(\Delta + \alpha + \Delta\alpha)m^0 \tag{189}$$

$$\hat{m}^t = \frac{\Delta}{2}\left(\frac{\Delta\alpha}{2}\right)^{t-1} \sum_{s=0}^{t-1}\left(\frac{\Delta\alpha}{2}\right)^{-s} \hat{m}^s + \frac{\alpha}{2}m^{t-1} \quad (t \geq 1) \tag{190}$$

$$m^t = \frac{\Delta^2\alpha}{4}\left(\frac{\Delta}{2}(1+\alpha)\right)^{t-2} \sum_{s=0}^{t-2}\left(\frac{\Delta}{2}(1+\alpha)\right)^{-s} m^s + \hat{m}^t + \frac{\Delta\alpha}{2}m^{t-1} \quad (t \geq 2) \tag{191}$$

$$R^{s0} = \hat{\chi}^{0s} \tag{192}$$

$$R^{st} = \hat{\chi}^{\min\{s,t\},\max\{s,t\}} - \sum_{l=0}^{t-1} \hat{Q}^{lt} R^{sl} \quad (t \geq 1) \tag{193}$$

$$Q^{0t} = \hat{m}^0 m^t + R^{0t} \tag{194}$$

$$Q^{st} = \hat{m}^s m^t + R^{st} + \frac{\Delta^2\alpha}{4}\left(\frac{\Delta}{2}(1+\alpha)\right)^{s-2} \sum_{l=0}^{s-2}\left(\frac{\Delta}{2}(1+\alpha)\right)^{-l} Q^{lt} + \frac{\Delta\alpha}{2}Q^{s-1,t} \quad (t \geq s \geq 1) \tag{195}$$

$$\hat{\chi}^{00} = \frac{\Delta\alpha}{4} \tag{196}$$

$$\hat{\chi}^{0t} = \frac{\Delta}{2}\left(\sum_{s=0}^{t-1}\chi^{s,t-1}\hat{\chi}^{0s} + \frac{\alpha}{2}(1-2\theta)m^{t-1}\right) \quad (t \geq 1) \tag{197}$$

$$\hat{\chi}^{st} = -\frac{\Delta}{2}\left(\sum_{l=0}^{t-1}Q^{\min\{s-1,l\},\max\{s-1,l\}}\hat{Q}^{lt} - \sum_{l=0}^{s-1}\chi^{l,s-1}\hat{\chi}^{lt} - m^{s-1}\hat{m}^t\right) \quad (t \geq s \geq 1). \tag{198}$$

We now consider the solution to these recurrence relations for sufficiently large $t$ ($t \gg 1$). Let us propose a trial solution of the form $m^t = c\mathcal{M}^t, \hat{m}^t = cL\mathcal{M}^t$, where $\mathcal{M} > \Delta(1+\alpha)/2$ and $c$ is a constant depending on the initial condition $\theta$. Note that $t$ in the left-hand side denotes the step number, while $t$ in the right-hand side is an exponent. We have:

$$\sum_{s=0}^{t-1}\left(\frac{\Delta\alpha}{2}\right)^{-s}\hat{m}^s = c\sum_{s=0}^{t-1}\left(\frac{\Delta\alpha}{2}\right)^{-s}L\mathcal{M}^s + \mathcal{O}(1) \tag{199}$$

$$\sum_{s=0}^{t-2}\left(\frac{\Delta}{2}(1+\alpha)\right)^{-s}m^s = c\sum_{s=0}^{t-2}\left(\frac{\Delta}{2}(1+\alpha)\right)^{-s}\mathcal{M}^s + \mathcal{O}(1) \tag{200}$$

From equations (190) and (191), we obtain the solution satisfying the condition $\mathcal{M} > \Delta(1+\alpha)/2$:

$$m^t = c(1+\Delta)\mathcal{M}^t \tag{201}$$

$$\hat{m}^t = c\mathcal{M}^t \tag{202}$$

where

$$\mathcal{M} = \frac{1}{4}\left(2\alpha\Delta + \alpha + \Delta + \sqrt{\alpha^2 + 2\alpha\Delta(2\Delta+1) + \Delta^2}\right). \tag{203}$$

Next, we consider solutions of the form $Q^{st} = c^2 q\mathcal{M}^{s+t}, R^{st} = c^2 r\mathcal{M}^{s+t}, \hat{\chi}^{st} = c^2\chi\mathcal{M}^{s+t}$ for $s, t \gg 1$. Substituting these into (193), we obtain

$$r = (1+\Delta)\chi \tag{204}$$

$$q = (1+\Delta)^2(1+\chi) \tag{205}$$

$$\chi = \frac{\Delta}{\alpha}\left(1 + \frac{\Delta}{(1+\Delta)^2}q\right). \tag{206}$$

Solving these equations yields

$$r = \frac{\Delta(1+\Delta)^2}{\alpha - \Delta^2}, \quad q = \frac{(1+\Delta)^2(\alpha+\Delta)}{\alpha - \Delta^2}, \quad \chi = \frac{\Delta(1+\Delta)}{\alpha - \Delta^2}. \tag{207}$$

Consequently, the generalization error as $t \to \infty$ is given by

$$H\left(\frac{m^t}{\sqrt{\Delta Q^{tt}}}\right) \to H\left(\sqrt{\frac{\alpha - \Delta^2}{\Delta(\alpha + \Delta)}}\right). \tag{208}$$

However, when $\alpha < \Delta^2$, this solution becomes inappropriate as $Q^{tt} < 0$. In this case, the scale $\mathcal{N}$ of $Q^{st} = c^2 q\mathcal{N}^{s+t}$ satisfies $\mathcal{N} > \mathcal{M}$, and the generalization error becomes $H(0) = 0.5$. This completes the proof of Result 3.

Result 3 reveals a phase transition phenomena at $\alpha = \Delta^2$. A generalization error of 0.5 means that the performance of $t$-SD is equivalent to random guessing; hence we refer to the phase $\alpha < \Delta^2$ as the performance collapse phase. The independence from the choice of the temperature is natural since it only affects on the scale of the weight in the linear $t$-SD model. In Figure 8, dependence of $\mathcal{E}^t$ on $\alpha$ and $\Delta$ at $t \to \infty$ is shown, with phase transition boundary represented by dashed line. The generalization error at $t \to \infty$ for $\alpha \geq \Delta^2$ is below 0.5, indicating performance better than random guessing, but it remains higher than the optimal error; hence, we refer to the phase $\alpha \geq \Delta^2$ as the intermediate performance phase.

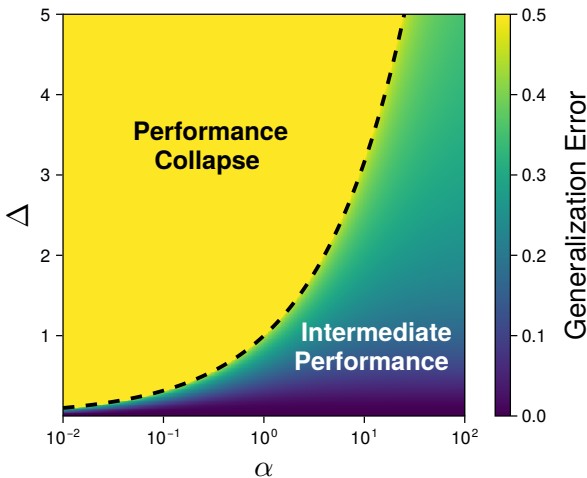

Figure 8: Theoretical prediction of generalization error for the linear $t$-SD model with $\lambda^0, \cdots, \lambda^t \to \infty$ and $t \to \infty$ with the phase transition boundary indicated by the dashed line.

## G   Experimental Details

This appendix provides the detailed experimental settings employed in Section 8. All experiments were executed on CPU workers equipped with an AMD EPYC 9654 processor and 512 GB of main memory. Each run required approximately two hours of wall-clock time using all available cores.

### G.1   Data and backbone selection

We consider the binary 'cat vs. dog' subset of CIFAR-10 Krizhevsky et al. [2009] (licensed under the MIT License) and employ two deep backbones, ResNet-18 and ResNet-50, pre-trained on ImageNet.

### G.2   Feature Extraction

Each CIFAR-10 image is first resized and normalized to match the preprocessing used during ImageNet training. We then remove the backbone's final classification layer and take the output of the penultimate layer as a fixed embedding of dimension $N = 512$ for ResNet-18 or $N = 2048$ for ResNet-50 maintainers and contributors [2016] (licensed under BSD 3-Clause "New" License). All embeddings and their clean labels are saved for downstream use.

### G.3   Label noise injection

To simulate noisy supervision, we flip each training label independently with probability $\theta$. Test labels remain untouched.

### G.4   Training subset sampling

From the pool of noisy embeddings, we uniformly draw $M$ samples (with a fixed class balance when desired) to form the actual training set used in the SD experiments.

### G.5   Self-distillation and hyperparameter tuning.

Using those $M$ examples, we perform the logistic $t$-SD procedure defined in Section 3. The key hyperparameters, $\lambda$ and $\beta$ are selected by minimizing the estimated generalization error on the test embeddings via Bayesian optimization. Concretely, we model the test error as a function of $(\lambda, \beta)$ with a Gaussian-process surrogate and optimize its expected improvement.

# H  Additional experiments

## H.1  Self-distillation on a noiseless dataset

We repeat our 1-SD experiments with no label noise ($\theta = 0$) to isolate the denoising effect from the dark knowledge effect. Since there is no label noise, any gain in generalization must arise solely from the teacher's soft outputs. As shown in Figure 9, no meaningful improvement is observed under realistic settings.

These findings validate the hypothesis of Section 5: in linear models under the Gaussian mixture model, dark knowledge alone yields only marginal benefit. The dominant mechanism by which SD enhances performance may be denoising, not the transmission of refined probability information even in more realistic scenarios.

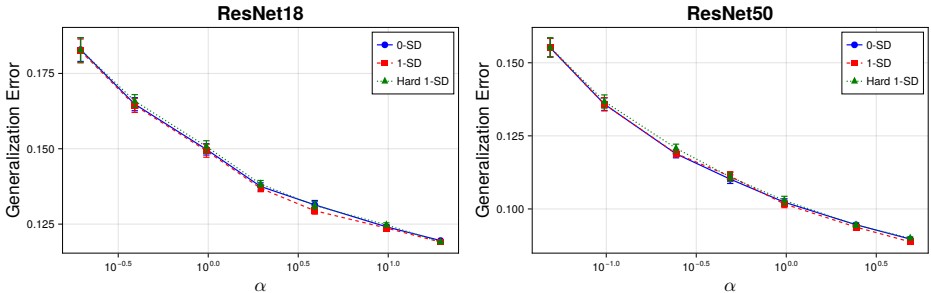

Figure 9: Comparison of the optimal generalization error of the logistic 0-SD model, 1-SD model and 1-SD model using hard pseudo labels for CIFAR-10 dog versus cat classification using pretrained ResNet-18 ($N = 512$) and ResNet-50 ($N = 2048$) feature representations. Parameters: $\theta = 0.0$. Error bars represent the standard error of the mean over 10 trials per point.

## H.2  Optimal soft labels for self-distillation

A natural question is whether the soft labels produced by the optimal 1-SD teacher simply mirror the probabilities predicted by the optimal 0-SD model, or whether they differ in a systematic way. Figure 10 shows, for several randomly chosen training samples, the ground-truth label, the noisy observed label, the predicted probability under optimal 0-SD, the pseudo-label assigned by the optimal 1-SD teacher, and the student's prediction under optimal 1-SD. The optimal teacher consistently issues more extreme confidence scores than the base model. This observation suggests that the most effective labels for student improvement can deviate from the model's optimal activations.

# I  Further remarks on limitations and future works

In this appendix, we outline several promising avenues that address the limitations of our current study and extend its insights.

Our theoretical analysis assumes purely linear models under Gaussian-mixture noise and relies on asymptotic $N, M \to \infty$ formulas, so its accuracy may degrade on finite-sample, non-linear settings. Likewise, although our CIFAR-10 probes demonstrate feasibility, they do not guarantee performance on larger or more complex vision tasks. Addressing these gaps suggests following several natural directions for further work.

## I.1  Extension to anisotropic data distributions

In anisotropic settings, the impact of SD is inherently direction-dependent: it tends to amplify useful signals along high-variance directions, while potentially neglecting or even distorting low-variance ones. As a consequence, the overall benefit of SD may become larger when the task-relevant information aligns with principal components, but smaller when crucial information lies in weak directions. Understanding this interplay between pseudo-label dynamics and the spectral structure of the data covariance is an important direction for future work.

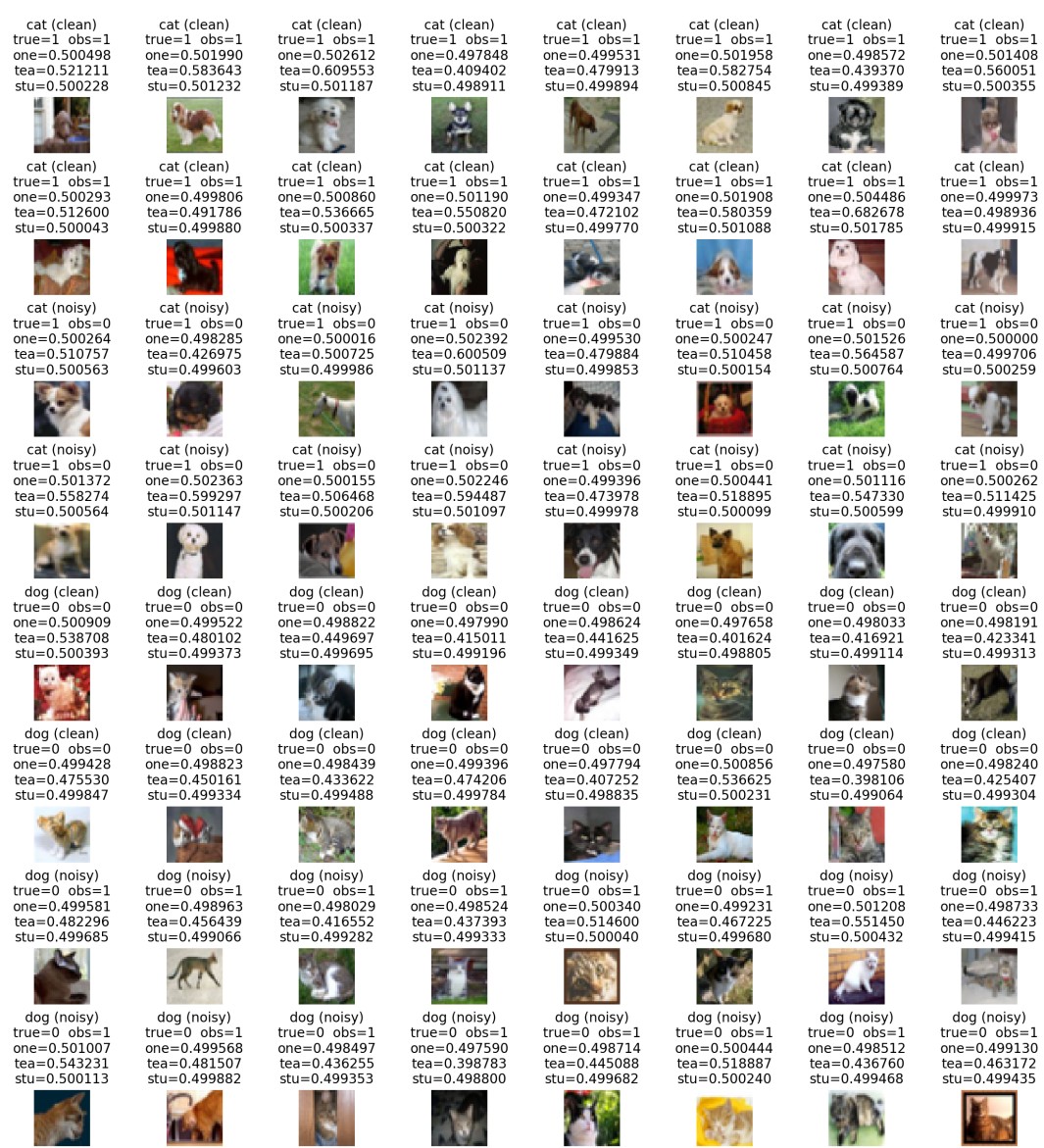

Figure 10: Sample images from the CIFAR-10 training set used in our experiments. 'true' and 'obs' indicate the ground-truth and noisy observed labels (0 for dog, 1 for cat), respectively. 'one', 'tea', and 'stu' denote the predicted labels under the optimal 0-SD model, the pseudo-labels provided to the student by the optimal 1-SD teacher, and the predicted labels under the optimal 1-SD student, respectively. Parameters: $\theta = 0.4, M = 1000, N = 512$; ResNet-18 features and logistic $t$-SD.

## I.2 Extension to other distillation strategies

While this study focuses on linear models, extending the analysis to deep learning presents promising directions for future research. In deep models, dark knowledge may differ significantly and hold greater significance than in linear analysis, due to their feature learning capabilities. For instance, models propagating intermediate layer information Zhang et al. [2019] might depend more on transfer of feature representations rather than predictions alone. Additionally, another avenue lies in exploring the interplay between SD and security, particuly optimizing defense against model stealing attacks Ma et al. [2021], Yilmaz and Keles [2025]. This line of inquiry extends our problem setting to a min-max framework, aiming to minimize the effectiveness of SD. Advancing these directions could contribute to both KD robustness and secure machine learning.

### I.3 What are the best pseudo-labels to learn?

Because the optimization goal of $t$-SD is the generalization error of the final ($t$-th) student, every intermediary teacher must issue labels not to mirror the true decision boundary but to maximize the downstream student's performance. Intriguingly, these student-centric labels differ substantially from the teacher-centric labels that would be chosen for standard classification, and more confident soft labels tend to produce stronger students. A more detailed analysis of these label distinctions is likely to reveal the answer to the fundamental question of which labels are most beneficial for learning.

### I.4 Application of multi-stage replica theory to other learning problems

We believe that the multi-stage replica method we employed for theoretical analysis can be extremely useful for probing learning dynamics that have, to date, remained inaccessible to DMFT. Problem formulations in which the outcomes of one learning phase are reused in a subsequent phase setting ups to which our framework can be applied directly include domain adaptation Blitzer et al. [2006], curriculum learning Bengio et al. [2009] and meta-learning Hospedales et al. [2022]. It would also be intriguing to investigate the generalization-error dynamics of deep neural networks under a regime where layers are trained sequentially and iteratively Cui et al. [2024].

