# OpenReview forum: "The Effect of Optimal Self-Distillation in Noisy Gaussian Mixture Model"
_NeurIPS.cc/2025/Conference — NeurIPS 2025 poster_

### Official Review · Reviewer_q2Aw · 2025-06-25

**Clarity:** 3
**Significance:** 3
**Originality:** 3
**Rating:** 4
**Confidence:** 3

**Summary:**

This paper explores the effectiveness of multi-stage self distillation based on linear classifier hyperparameter optimization in binary classification tasks of noisy Gaussian mixture data using the replica method in statistical physics. The main driving force for the improvement of self distillation (SD) performance was found to be the denoising effect achieved through hard pseudo labeling, and two practical heuristic strategies were proposed to enhance SD performance.

**Questions:**

1. Only exploring binary classification tasks, the generalization of this work needs to be verified.
2. Is the CIFAR-10 dataset validated on a larger dataset as the image resolution is relatively small? For example, ImageNet.

**Ethical Concerns:**

["NO or VERY MINOR ethics concerns only"]

**Final Justification:**

Thank you for the author's clarification. I have clarified the previous questions and will maintain my score.

**Limitations:**

yes.

**Paper Formatting Concerns:**

None.

**Quality:**

2

**Strengths And Weaknesses:**

Strengths：1. This method provides an environment that can simultaneously analyze the "dark knowledge" hypothesis and the denoising hypothesis, and analyze both.
2. Theoretical analysis and experimental verification are complete.

Weaknesses：
1. Only exploring binary classification tasks, the generalization of this work needs to be verified.
2. Is the CIFAR-10 dataset validated on a larger dataset as the image resolution is relatively small? For example, ImageNet.

---

> ### Author Rebuttal · Authors · 2025-07-29
>
> We thank you for your valuable time and insightful feedback. We address specific comments below.
>
> **Reply to the Questions:**
>
> > 1. Only exploring binary classification tasks, the generalization of this work needs to be verified.
>
> We hypothesize that in the multi-class setting, the importance of dark knowledge would become even more pronounced. As the output dimension increases, the richer information contained in the multi-dimensional probability distribution of the logits would likely play a greater role.
> For this reason, we believe that extending our analysis to multi-class classification is a crucial and exciting direction for future research. For this, the present result would provide the theoretical framework necessary for tackling this important extension.
>
> > 2. Is the CIFAR-10 dataset validated on a larger dataset as the image resolution is relatively small? For example, ImageNet.
>
> The primary goal of our experiments was to provide a qualitative validation of our theoretical findings in a real-world data setting, rather than to achieve state-of-the-art performance. CIFAR-10 is a standard and well-understood benchmark that is sufficient to demonstrate that the findings obtained from our theory hold true in practice. We agree that validation on larger-scale datasets like ImageNet is an important direction for future work.

---

### Official Review · Reviewer_qzrp · 2025-06-29

**Clarity:** 2
**Significance:** 1
**Originality:** 2
**Rating:** 3
**Confidence:** 4

**Summary:**

This paper studies the self-distillation technique on the binary classification problem on GMMs with corruption in the labels in the asymptotic regime and focuses on the generalization error of linear classifiers obtained through ridge regression and $\ell_2$-regularized cross-entropy loss. By employing the non-rigorous replica trick, precise expressions characterizing the generalization error are provided through self-consistent nonlinear equations. Using these derivations the utility of the dark knowledge and the effects of soft/hard labels in self-distillation across different dataset sizes and noise levels are investigated. Finally, it is claimed that naively applying self-distillation will deteriorate the performance, and a heuristic for appropriately learning the bias terms in the unbalanced setting is provided.

**Questions:**

1. Some comments:

    **1.1.** I could not find the definition for the term "pseudo-label" in the abstract (line 8).

    **1.2.** Does the definition of the temperature parameter only apply for the sigmoid and logistic loss in line 146?

    **1.3.** I could not find the definition of $\Delta$ in the main body and it is only defined in Appendix B.1.

    **1.4.** The sentence "These findings suggest that, at least within our toy model, dark knowledge is not the key factor driving the success of SD" in line 70 is in contrast with the sentence in line 213.

    **1.5.** What does "diversity of teacher prediction" mean in line 281?


**2.** In Figure 2A, is there an optimal corruption level in the overparametrized regime ($\alpha<1$)?


**3.** In Section 7, how does regularizing the bias term affect the performance?

**Ethical Concerns:**

["NO or VERY MINOR ethics concerns only"]

**Final Justification:**

I would like to thank the authors for their responses. The authors have adequately addressed the questions 1-3 raised in my review.  However, I remain unconvinced about some of the points outlined in the Weaknesses section (points 3.1 and 3.2):

1. The assumptions on the class means being antipodal and the covariances being isotropic are too restrictive.

2. The authors did not respond to my latest comment regarding whether their proposed early stopping strategy is a sensible approach.

In the light of this discussion, I would like to maintain my score.

**Limitations:**

Yes.

**Quality:**

2

**Strengths And Weaknesses:**

**Strengths**

**1.** By considering self-distillation, an important paradigm in training foundational models, this work has provided theoretical insights into when SD can prove useful and what its limitations are.

**2.** The authors have provided extensive discussion in Sections 5-7 around Result 4.1 to make the derived expressions interpretable. In the single-step case, they investigated the efficacy of soft and hard labels for various noise levels, corruption levels, and overparametrization ratios. The paper demonstrates that depending on the dataset size, the improvement in the performance through self-distillation may be marginal in the multi-stage setting.

**3.** The problem setting and the notation are defined clearly, and the paper is well written in general.

**Weaknesses**

1. The authors report their findings through the non-rigorous replica method as "Theorem". I believe that the title of statements should be changed from "Theorem" to "Result" or "Finding" to reflect the lack of rigor. It is recommended to use the word "Computation" instead of the word "Proof". There are also other instances in the paper which require rephrasing to indicate that the approach is non-rigorous.  Adding a discussion on potential approaches to proving the derived expressions will improve the quality of the paper.

2.  The literature review in the paper could be more comprehensive:

    **2.1.**  Starting from the introduction, referring to more recent papers using self-distillation in practice motivates problem formulation further as the currently cited papers in lines 28-29 are from 2019. Is self-distillation being used actively? How close is the formulation of Self-Distillation technique considered in this paper to what is being employed in practice? I would appreciate authors' comments on how applications motivate the current problem setting.

    **2.2.** In lines 58-59, the authors mention CGMT and Gordon's inequality as separate approaches for analyzing high-dimensional inference problems in the asymptotic regime. CGMT relies on Gordon's inequality to provide precise results in the asymptotic regime. Citing Gordon's inequality as a separate approach is confusing. I would appreciate the author's comment on this. Furthermore, I believe that adding additional references on the asymptotics of linear binary classification and providing a brief overview of known results will improve the exposition. Related works on the asymptotics of GMM classification are as follows : [1], [2], [3], [4]

    **2.3.** [3] rigorously analyzed the linear binary classification problem using ridge regression and other regularizers in the presence of label noise (corruption). I recommend referring to this paper in the paragraph "Fixed point analysis and learning dynamics" in line 255 as [3] also focuses on the effect of corruption and the large $\lambda$ regime.

3. While I appreciate the extensive discussion following Result 4.1, I believe the current result is limited in scope, given the theoretical nature of the work. As shown in [3] (Remark 2), taking the regularization strength to be large recovers the oracle optimal classifier in the balanced setting for arbitrary corruption level $< 0.5$. This challenges the need for successive use of self-distillation and training the model from scratch for each $t$; whereas by taking the regularization strength to be large and training only once yields the optimal performance. I would appreciate the authors' comment on this.

    **3.1.** Regarding the early-stopping heuristic in line 72 and Result 6.1, I am not convinced that only considering the setting $\lambda_t \rightarrow \infty$ would suffice to claim that early-stopping is a sensible strategy.

    **3.2.** Potential tangential directions to complement the results include performing the computations for a more general mean and covariance structure as it is not immediate what the optimal performance of the ridge regression would be for arbitrary covariance matrices.

    **3.3.** Furthermore, characterizing the evolution of the optimal regularization strength through self-consistent equations and the generalization error can further strengthen Result 6.1.

    **3.4.** The discussions in Sections 5-7, despite being intuitive, lack the necessary mathematical details. I recommend presenting some of the mathematical expressions of the predicted generalization error from the Appendix in the main body (such as (158) and (192) in the Appendix). Discussing how the authors arrive at the observations in Section 5-7 by referring to the mathematical expressions adequately would help in understanding the computations.

4. Other comments:

    **4.1.** The overparametrizated regime ($\alpha < 1$), despite being fundamentally important, remains underexplored in this paper. For instance, in Figures 1 and 2, it is difficult to discern how the error behaves when $\alpha < 1$ varies.  I suggest adding separate figures in the spirit of Figures 1 and 2 for the overparametrized and underparametrized regimes.

   **4.2.** To support their proposed heuristic in Sections 6 and 7, the authors should conduct additional experiments on real-world dataset.

**[1]** Kini, Ganesh Ramachandra, and Christos Thrampoulidis. "Analytic study of double descent in binary classification: The impact of loss." 2020 IEEE International Symposium on Information Theory (ISIT). IEEE, 2020.

**[2]** Deng, Zeyu, Abla Kammoun, and Christos Thrampoulidis. "A model of double descent for high-dimensional binary linear classification." Information and Inference: A Journal of the IMA 11.2 (2022): 435-495.

**[3]** Akhtiamov, Danil, Reza Ghane, and Babak Hassibi. "Regularized linear regression for binary classification." 2024 IEEE International Symposium on Information Theory (ISIT). IEEE, 2024.

**[4]** Ghane, Reza, Danil Akhtiamov, and Babak Hassibi. "Universality in transfer learning for linear models." Advances in Neural Information Processing Systems 37 (2024): 125729-125779

---

> ### Author Rebuttal · Authors · 2025-07-29
>
> We thank you for your valuable time and insightful feedback. We address specific comments below.
> Please note that due to word count limitations, we may omit quotations from the original comments. We kindly ask you to refer to the original review for the complete questions.
>
> **Comments on the weakness:**
>
> > 1. The authors report their findings through the non-rigorous replica method as "Theorem". I believe…
>
> Our results are derived from the replica method under the Replica Symmetric (RS) assumption. For the convex problem we consider, predictions based on the RS assumption are strongly believed to be exact. Therefore, while our findings are not yet rigorously proven in a mathematical sense, they are considered to be solid predictions within the statistical physics community. Our initial choice to use "Theorem" was intended to clarify the paper's structure and highlight our main findings, a convention also seen in other related works [1, 2]. However, we acknowledge your point that this term could be misleading. We will revise the wording accordingly in the camera-ready version.
>
> > The literature review in the paper could be more comprehensive:
> > **2.1.** Starting from the introduction, referring to more recent papers using self-distillation in practice motivates problem formulation further as the currently cited papers in lines 28-29 are from 2019. Is self-distillation being used actively? How close is the formulation of Self-Distillation technique considered in this paper to what is being employed in practice? I would appreciate authors' comments on how applications motivate the current problem setting.
>
> SD is indeed actively applied to practical problems in recent studies [3, 4], with current research often focusing on specialized methods like propagating internal representations. **However, our focus—propagating the output—also remains a practical and current research topic [5, 6], and we believe our work offers significant insights for development of modern algorithms. Furthermore, our mathematical model serves as a valuable foundation for the theoretical analysis of mainstream contemporary SD methods, including those that propagate internal representations [3].**  The potential for such extensions to more modern methods is also discussed in Appendix I.
>
> > **2.2.** In lines 58-59, the authors mention CGMT and Gordon's inequality as separate approaches for analyzing high-dimensional inference problems in the asymptotic regime. CGMT relies on Gordon's inequality…
>
> Thank you for your advice and giving additional reference. We agree that presenting CGMT and Gordon's method separately was confusing. We will fix this point. Also, we will add to and revise the manuscript based on these for the camera-ready version.
>
> > **2.3.** [3] rigorously analyzed the linear binary classification problem using ridge regression and other regularizers in the presence of label noise (corruption). I recommend…
>
> Thank you for your advice and giving additional reference. We were not aware that it was valid for such a general setting. We plan to add to and revise the manuscript based on these for the camera-ready version.
>
> > 3. While I appreciate the extensive discussion following Result 4.1, I believe the current result is limited in scope, given the theoretical nature of the work. As shown in [3] (Remark 2), taking the regularization strength to be large recovers the oracle optimal classifier in the balanced setting for arbitrary corruption level . This challenges the need for successive use of self-distillation and training the model from scratch for each t ; whereas by taking the regularization strength to be large and training only once yields the optimal performance. I would appreciate the authors' comment on this.
>
> It is sure that $\lambda_0 \to \infty$ leads to the Bayes optimal classifier. However, the situation changes after 1-SD because the labels used have different nature from the original labels. This is the non-trivial aspect of SD that should be investigated in detail.
> **Our results show that SD can further improve performance beyond the initial Bayes optimal estimation of the standard supervised setup (referred to as "optimal 0-SD" in our notation)**, as seen for example in Fig 3A. Therefore, applying SD remains a valuable approach in the current setting.
>
> > **3.1.** Regarding the early-stopping heuristic in line 72 and Result 6.1, I am not convinced that only considering the setting $\lambda_t \to \infty$ would suffice to claim that early-stopping is a sensible strategy.
>
> While we did not include it in the current draft, we have confirmed in our linear model setting that the optimal hyperparameter schedule, derived from the black-box optimization, follows the following pattern: it begins with a very large regularization ($\lambda\to\infty$) for an initial period and then switches to zero regularization $\lambda\to 0$ thereafter.
> This is precisely what motivates our claim that early stopping is a sensible strategy through the $\lambda_t \to \infty$ hyper-parameter schedule. To provide full justification for this claim, we will add the detailed results of this optimal hyper-parameter schedule to the appendix in the camera-ready version.
>
> > **3.2.** Potential tangential directions to complement the results include performing the computations for a more general mean and covariance structure as it is not immediate what the optimal performance of the ridge regression would be for arbitrary covariance matrices.
>
> We agree that analyzing the model with a more general covariance structure is an　interesting extension. We will add this as a key direction for future work.
>
> > **3.3.** Furthermore, characterizing the evolution of the optimal regularization strength through self-consistent equations and the generalization error can further strengthen Result 6.1.
>
> See the reply to question 3.1.
>
> > **3.4.** The discussions in Sections 5-7, despite being intuitive, lack the necessary mathematical details. I recommend presenting some of the mathematical expressions of the predicted generalization error from the Appendix in the main body (such as (158) and (192) in the Appendix). Discussing how the authors arrive at the observations in Section 5-7 by referring to the mathematical expressions adequately would help in understanding the computations.
>
> Thank you for your advice. Generally, it is not feasible to write a simple, closed-form expression for the generalization error. For this reason, we have validated our theoretical results with numerical calculations, which is one of the standard practices in asymptotic analysis. Furthermore, we wish to prioritize an intuitive presentation for a broad readership. In addition, the expression for the generalization error is given in Eq. (9), and we believe the necessary mathematical expressions in the main body are self-contained.
>
> **Reply to the Questions:**
>
> >  **1.1.** I could not find the definition for the term "pseudo-label" in the abstract (line 8).
>
> The detailed definition of pseudo-label is described in line 144-146. We will clarify this by adding the following definition in the camera-ready version: "A pseudo-label is an artificially generated label added to the training set to retrain and improve the model." to avoid any misunderstanding.
>
> > **1.2.** Does the definition of the temperature parameter only apply for the sigmoid and logistic loss in line 146?
>
> Yes, as written in line 153-154. However, we will add an explanation to avoid any misunderstanding in the camera-ready version.
>
> > **1.3.** I could not find the definition of  $\Delta$ in the main body and it is only defined in Appendix B.1.
>
> The $\Delta$ is the size of each cluster, as defined in the right hand side of Eq.(1).
>
> >  **1.4.** The sentence "These findings suggest that, at least within our toy model, dark knowledge is not the key factor driving the success of SD" in line 70 is in contrast with the sentence in line 213.
>
> The statement in line 70 argues that the **absolute effect** of dark knowledge is small. In contrast, the statement in line 213 argues that the **relative effect** of dark knowledge (from soft labels) is large compared to that of hard labels. While these two statements are consistent, we agree the phrasing could be confusing. We will add a clarification to avoid any misunderstanding in the camera-ready version.
>
> > **1.5.** What does "diversity of teacher prediction" mean in line 281?
>
> Diversity is a measure of the population spread of a teacher model's predictive probabilities across different samples in a dataset [7].  We insist this concept is corresponding to our "predictive uncertainty". We will add this explanation to a camera-ready version.
>
> > **2.** In Figure 2A, is there an optimal corruption level in the overparametrized regime ($\alpha < 1$)?
>
> We believe the answer is no. The apparent peak is likely an artifact of numerical instability in our black-box hyperparameter optimization, which becomes particularly challenging in that regime.
>
> > **3.** In Section 7, how does regularizing the bias term affect the performance?
>
> Regularizing the bias term is tempting to balance the magnitude with the weight. However, we do not investigate the bias-regularized setting since we believe it is not a general setting.
>
> [1] Barbier et al., _PNAS_ **116**, 5451 (2019). \
> [2] Cornacchia et al., _Mach. Learn.: Sci. Technol._ **4**, 015019 (2023). \
> [3] Yang et al., arXiv:2402.13669 (2024). \
> [4] Chen et al., arXiv:2505.21501 (2025).\
> [5] Yilmaz and Keles, arXiv:2403.05181 (2024). \
> [6] Ma et al., _ICLR_ (2021). \
> [7] Zhang and Sabuncu, _NeurIPS_ (2020).

---

> > ### Comment · Reviewer_qzrp · 2025-08-08
> >
> > I appreciate the authors' response to my questions.
> >
> > > While we did not include it in the current draft, we have confirmed in our linear model setting that the optimal hyperparameter schedule, derived from the black-box optimization, follows the following pattern: it begins with a very large regularization ($\lambda\to\infty$) for an initial period and then switches to zero regularization $\lambda\to 0$ thereafter. This is precisely what motivates our claim that early stopping is a sensible strategy through the $\lambda_t \to \infty$ hyper-parameter schedule.
> >
> > In the overparametrized regime, the classifier obtained through $\lambda \to 0$ for the mse loss corresponds to the minimum $\ell_2$ norm interpolating solution. Furthermore, we know from the implicit bias property of GD, this solution is also the point of convergence for GD when initialized from zero. In the light of the claim above, I am not convinced taking $\lambda_t \to \infty$ is a sensible strategy as large regularization is not a practical approach in many applications; Additionally, optimal $\lambda_t \to 0$ implies that the effect of corruption in the labels is mitigated and one can interpolate the labels without loss of performance, which could be done by running gradient descent.

---

### Official Review · Reviewer_7Ees · 2025-07-01

**Clarity:** 2
**Significance:** 3
**Originality:** 3
**Rating:** 4
**Confidence:** 3

**Summary:**

This paper presents a theoretical and empirical study of multi-stage self-distillation for binary classification under label noise. Using the replica method from statistical physics, the authors derive exact asymptotic expressions for the generalization error of a linear classifier trained via an optimally-tuned SD schedule on a noisy Gaussian-mixture model. The analysis shows that the dominant benefit of SD arises from a denoising effect of (hard) pseudo-labels, while the additional “dark-knowledge’’ contained in soft probabilities plays only a marginal role. Two practical heuristics naturally emerge from the theory: (i) early stopping after a few distillation stages to avoid performance collapse, and (ii) bias fixing in imbalanced-label regimes. Both predictions are confirmed experimentally on CIFAR-10 (cat vs dog) using frozen ResNe backbones, where distillation yields an error reduction at medium data sizes and behaves as predicted by the theory.

**Questions:**

* Please clarify the logical status of Theorem 4.1 (and others) is it a conjecture supported by extensive evidence, or is there a rigorous proof in this particular Gaussian-mixture/linear-classifier regime?  If it is still conjectural, could the authors outline a plausible proof strategy (e.g. AMP/state-evolution or convex Gaussian min-max techniques) and discuss outstanding technical obstacles?  Conversely, if a proof does exist (even for a restricted loss/activation setting), please cite or sketch it explicitly, so readers can distinguish rigorously established results from physics-style predictions.  To be clear, I think it is ok to use the replica method, but statement like "theorem" should be used precisely. There are a number of rigorous predictions, that the authors cite so I was wondering... for instance,

* Extension beyond linear models: Could the authors discuss how the order-parameter decomposition might generalize to shallow networks with learned features, or suggest what additional assumptions would be required?

* Computational cost of exhaustive tuning: The theory assumes hyper-parameters are chosen to minimize the asymptotic error. How sensitive are the results to imperfect tuning in practice, and can the replica formula guide efficient hyper-parameter search?

*Robustness to non-random (adversarial) label noise: Have the authors investigated whether the denoising advantage persists under structured or instance-dependent noise?

* Scaling of the early-stopping heuristic: The stopping rule is motivated by the peak of the signal term . Could the authors propose a data-driven criterion that does not rely on inaccessible order parameters?

* Broader experimental validation: The CIFAR-10 cat-vs-dog split is convenient for binary evaluation; do the authors expect similar gains on multi-class tasks such as CIFAR-100 or ImageNet subsets? Evidence here could strengthen the empirical story.

**Ethical Concerns:**

["NO or VERY MINOR ethics concerns only"]

**Final Justification:**

This paper provides a novel and insightful theoretical treatment of multi-stage self-distillation under label noise using the replica method, complemented by experiments that qualitatively confirm its predictions. The rebuttal was detailed and transparent, clarifying the logical status of the replica-based results, the obstacles to rigorous proofs, and the practical scope of the heuristics. While the analysis remains confined to linear Gaussian models with limited empirical validation, the work offers valuable conceptual clarity on why and when self-distillation helps. These strengths outweigh the limitations, and I therefore maintain my borderline-accept recommendation.

**Limitations:**

yes

**Quality:**

3

**Strengths And Weaknesses:**

The paper’s primary strength lies in its systematic analytical framework for studying multi-stage self-distillation under label noise: by wielding the replica method, the authors derive closed-form asymptotic expressions for the generalization error of a linear classifier, giving the community a rare, almost “microscopic” view of how successive distillation stages denoise hard pseudo-labels and why soft-label “dark knowledge” contributes comparatively little. This theoretical clarity is complemented by carefully designed CIFAR-10 experiments that track the predicted order parameters and reproduce all qualitative trends—most notably the early-stopping sweet spot and the bias-correction benefit in imbalanced regimes—lending empirical credibility to the analysis. Beyond these immediate results, the work is significant because it reframes a widely used heuristic (self-distillation) as an optimal-control problem, yielding actionable guidance: practitioners can tune the number of stages, stop training when the signal term peaks, and apply bias fixing to counter class imbalance, all with principled justification. The presentation is generally lucid: major claims are boxed, proofs (albeit non-rigorous) are deferred to well-labeled appendices, and the code release fosters reproducibility. From an originality standpoint, extending replica calculations—traditionally applied to single-shot learning or convex optimization—to a sequential learning protocol and extracting concrete training heuristics is, to the best of my knowledge, novel.

 Nevertheless, weaknesses remain. Most conspicuously, the theoretical guarantees are confined to linear models trained on Gaussian-mixture data; while this stylized setting captures the essence of label noise, it leaves open the quantitative fidelity of the formulas when one moves to deep, finite-data regimes. Relatedly, the empirical validation, though well executed, is narrow in scope: a single binary split of CIFAR-10 shows roughly a five-percentage-point error drop—a respectable but not game-changing improvement—and no large-scale or multi-class experiments are reported. Accessibility is another concern: the dense replica derivations, coupled with the replica-symmetry assumption, may deter readers unfamiliar with statistical-physics techniques, and the paper does not fully clarify whether its “Theorem 4.1” is rigorously proven or best viewed as a conjecture supported by evidence. Finally, while the authors note hyper-parameter tuning is guided by their closed-form error expression, they do not quantify how sensitive real-world gains are to imperfect tuning, nor do they assess robustness to structured or adversarial noise. In sum, the paper offers a well-crafted, theoretically insightful contribution with clear practical hooks, yet its impact could be amplified by broader empirical validation, a more explicit discussion of rigor versus heuristic derivation, and guidance on extending the framework beyond the linear-Gaussian sandbox.

---

> ### Author Rebuttal · Authors · 2025-07-29
>
> We thank you for your valuable time and insightful feedback. We address specific comments below.
>
> **Comments on the weakness:**
>
> > it leaves open the quantitative fidelity of the formulas when one moves to deep, finite-data regimes.
>
> Thank you for raising the concern regarding the realistic setups. As for the finite-data regime, we confirm in Appendix D that our predictions for the asymptotic limit are empirically consistent with numerical experiments of finite size, where the input dimension is on the order of 10^2. Therefore, we expect that the finite-size effects are not a major concern. As for the fidelity in deep cases, it is true that our results do not directly extend to such cases. Nonetheless, we suspect that some of the insights obtained in the paper may hold broadly. For further discussion on this point, please refer to our response to Reviewer F2RL.
>
> > Relatedly, the empirical validation, though well executed, is narrow in scope: a single binary split of CIFAR-10 shows roughly a five-percentage-point error drop—a respectable but not game-changing improvement—
>
> Our primary goal in this study is to obtain reliable insights into the mechanisms underlying self-distillation (SD), which can remain unclear based on limited empirical evidence. The CIFAR-10 experiment was intended as a sanity check to examine whether the key qualitative findings predicted by our theory also emerge in a more practical setting beyond linear classifiers. While the scope of these experiments is limited, the results are consistent with our theoretical predictions, which we believe supports the validity of our analysis. We agree that more extensive empirical validation on large-scale and diverse benchmarks would be an important direction, and we leave this for future work.
>
> > Accessibility is another concern: the dense replica derivations, coupled with the replica-symmetry assumption, may deter readers unfamiliar with statistical-physics techniques
>
> We acknowledge that the replica-based analysis inevitably involves mathematically dense derivations, which may present an accessibility challenge to readers unfamiliar with statistical physics. However, given the goal of precisely characterizing the behavior of the trained model,  we believe that a certain level of technical detail is intrinsically unavoidable. To address this, we defer the core derivations and technical steps in the appendix, keeping the main text focused on conceptual explanations and key results. We believe this structure allows readers to adjust the depth of engagement depending on their background and interests.
>
>
> **Other comments regarding the weakness overlap with the content of the question, so I will describe them below as answers to the question.**
>
>
> **Reply to the Questions:**
>
> > Please clarify the logical status of Theorem 4.1 (and others) is it a conjecture supported by extensive evidence, or is there a rigorous proof in this particular Gaussian-mixture/linear-classifier regime? If it is still conjectural, could the authors outline a plausible proof strategy (e.g. AMP/state-evolution or convex Gaussian min-max techniques) and discuss outstanding technical obstacles? Conversely, if a proof does exist (even for a restricted loss/activation setting), please cite or sketch it explicitly, so readers can distinguish rigorously established results from physics-style predictions. To be clear, I think it is ok to use the replica method, but statement like "theorem" should be used precisely. There are a number of rigorous predictions, that the authors cite so I was wondering... for instance,
>
>
> Our results are derived from the replica method under the Replica Symmetric (RS) assumption. Although formally proving the validity of the analytic continuation from integer replica number to real value is a long-standing mathematical problem, for the convex optimization we consider, predictions based on the RS assumption are widely believed to be exact. Therefore, while our findings are not yet rigorously proven in the strict mathematical sense, they are considered to be reliable predictions within the statistical physics community.
>
> Regarding a rigorous proof without the replica method, the presence of temporal correlations in our model presents a significant technical obstacle. For example, the AMP-based proof strategy to handle such a time-correlation used in [1, 2] is limited to the so-called first-order algorithms, which does not cover the current multi-stage SD procedure. Also, the application of CGMT for the dynamics, which could be applied to multi-stage SD, is often done in on-line settings [3], where a new fresh batch of data is used at each stage of optimization. Hence, formally proofing the current result would require developing new mathematical tools.
>
> Finally, we were unsure about the point you were beginning to make with "for instance,"​​, but we hope our explanation clarifies the status of our results.
>
> > Extension beyond linear models: Could the authors discuss how the order-parameter decomposition might generalize to shallow networks with learned features, or suggest what additional assumptions would be required?
>
> First, the primary contribution of our paper is to provide a precise, analytical solution for the linear SD model, which in itself was a significant and previously unaddressed challenge. We believe that establishing this rigorous baseline is a necessary first step before tackling more complex architectures.
> Regarding generalization, an extension might be possible under certain simplifying assumptions. For instance, our analytical framework could likely be adapted to a two-layer random feature model under one step SGD, as the analysis in that setup is similar to a random feature model with a spike signal.
> However, extending the analysis to a shallow network where all features are fully learned is a much more formidable challenge. This is an important open problem that is of great interest to the broader field, and it extends beyond the specific scope of our current work.
>
> >  Computational cost of exhaustive tuning: The theory assumes hyper-parameters are chosen to minimize the asymptotic error. How sensitive are the results to imperfect tuning in practice, and can the replica formula guide efficient hyper-parameter search?
>
> Based on our numerical results for the linear model, any sufficiently large value of the regularization parameter $\lambda$ yields near-optimal results. We also found that the temperature parameter has a minimal impact on the outcome. Consequently, the replica theory indicates that an exhaustive hyperparameter search is not critical for this model.
> How these sensitivities would change in more general and complex settings remains an open question for future investigation.
>
> > Robustness to non-random (adversarial) label noise: Have the authors investigated whether the denoising advantage persists under structured or instance-dependent noise?
>
> No; however, we agree that this is a valuable and interesting direction for research. There is a wide variety of ways to model structured noise, and a meaningful investigation would require carefully specifying the type of noise in a concrete context. We will add this to future work in the camera-ready version.
>
> > Scaling of the early-stopping heuristic: The stopping rule is motivated by the peak of the signal term . Could the authors propose a data-driven criterion that does not rely on inaccessible order parameters?
>
> Our results indicate that once performance drops, it does not recover. Therefore, we believe a practical and sufficient data-driven criterion is to simply stop training as soon as performance on a validation set starts to degrade.
>
>
> > Broader experimental validation: The CIFAR-10 cat-vs-dog split is convenient for binary evaluation; do the authors expect similar gains on multi-class tasks such as CIFAR-100 or ImageNet subsets? Evidence here could strengthen the empirical story.
>
> We hypothesize that in the multi-class setting, the importance of dark knowledge would become even more pronounced. As the output dimension grows, the richer information contained in the multi-dimensional probability distribution of the logits would likely play a larger role.
> For this reason, we believe that extending our analysis to multi-class classification is a crucial and exciting direction for future research. For this, the present result would provide a theoretical framework necessary for tackling this important extension.
>
>
>
>
>
>
> [1] Z. Fan. Approximate message passing algorithms for rotationally invariant matrices. The Annals of Statistics, 50(1):197–224, 2022.
>
> [2] S. Liu. and J. Ma. Unifying AMP Algorithms for Rotationally-Invariant Models. arXiv preprint arXiv:2412.01574.
>
> [3] K.A. Chandrasekher, A. Pananjady. and C. Thrampoulidis. Sharp global convergence guarantees for iterative nonconvex optimization with random data. The Annals of Statistics, 51(1), pp.179-210. 2023

---

> > ### Comment · Reviewer_7Ees · 2025-08-07
> >
> > I appreciate the authors’ detailed and transparent rebuttal, which clarified the logical status of their replica-based “theorems,” outlined the obstacles to a rigorous proof, and explained the practical scope of their heuristics. Although key extensions—especially to fully-trained deep networks and structured-noise settings—remain open, the current work offers a valuable and novel theoretical lens on self-distillation that, in my view, merits dissemination to the community. I am therefore inclined toward acceptance; however, in keeping with NeurIPS policy and to reflect the limitations, I will retain my original score.

---

### Official Review · Reviewer_F2RL · 2025-07-03

**Clarity:** 3
**Significance:** 2
**Originality:** 3
**Rating:** 5
**Confidence:** 2

**Summary:**

This paper demystifies self-distillation (SD), a puzzling yet effective machine learning technique where models bootstrap their own performance by learning from their previous outputs. While widely adopted, the mechanisms behind SD's success have remained elusive until now. The researchers tackle this knowledge gap through a rigorous theoretical analysis, applying statistical physics' replica method to examine multi-stage SD with linear classifiers operating on noisy Gaussian mixture datasets. Their investigation reveals a crucial insight: SD's effectiveness primarily stems from its noise-reduction capabilities when generating hard pseudo-labels, with particularly pronounced benefits in datasets of moderate size. The analysis yields two practical enhancements for practitioners: implementing stage-limiting early stopping procedures (which shows broad applicability), and stabilizing bias parameters when dealing with imbalanced label distributions. Moving beyond theory, the authors validate these findings through real-world experiments on CIFAR-10 using pretrained ResNet architectures. This work bridges the gap between theoretical understanding and practical application, providing both fundamental explanations and actionable strategies for leveraging SD in environments where noise poses significant challenges.

**Questions:**

- The authors of the paper identified to two practical heuristics to enhance self distillation. How do they perform when applied to more realistic scenarios (e.g. ResNet for CIFAR10/100 when the entire neural network is trainable)?

**Ethical Concerns:**

["NO or VERY MINOR ethics concerns only"]

**Final Justification:**

The authors argued that training full scale neural networks can be challenging during reviewer discussion. I still believe that training the full ResNet models on CIFAR datasets should be doable. Including the additional experiments on these could further enhance the paper. Nevertheless, I think the theoretical insights are valuable. As such, I am retaining my original score.

**Limitations:**

Yes

**Quality:**

3

**Strengths And Weaknesses:**

### Strengths
- The paper is well structured and well written.
- The problem studied is of general interest to lots of researchers in the field.
- The theoretical study was rigorously carried out and technically sound.
- The conclusions drawn were interesting and insightful.

### Weaknesses
- My major complaint is the disconnection between the simplified linear classification of Noisy Gaussian Mixture Model and deep neural networks that distillation has found success in. While the authors of the paper tried to supplement the theory with experiments on CIFAR-10 classification using pretrained ResNet backbone, the experiments are still quite different from the real-world setting that self-distillation is applied. Indeed, the entire neural network weights are typically trained during self distillation. It would be much more interesting if the authors of the paper can also show that the conclusions drawn with linear classifier can be observed in the more general and realistic scenario as well.

---

> ### Author Rebuttal · Authors · 2025-07-29
>
> We thank you for your valuable time and insightful feedback. We address specific comments below.
>
> **Reply to the Questions:**
>
> > The authors of the paper identified to two practical heuristics to enhance self distillation. How do they perform when applied to more realistic scenarios (e.g. ResNet for CIFAR10/100 when the entire neural network is trainable)?
>
> We appreciate the reviewer's interest in the applicability of our heuristics to more realistic scenarios, which are indeed practically important.
>
> Unfortunately, conducting large-scale experiments with fully trainable networks is challenging due to the significant effort required for hyperparameter tuning. However, our analysis of the linear model may provide a strong theoretical foundation to discuss the potential utility and outlook for these heuristics in more complex scenarios. Below, we describe our current outlook on the utility of each heuristics.
>
> Early Stopping: Our results for the linear model indicate that once performance begins to degrade, it does not recover with further training (as seen in Figure 3C). If we assume this monotonic degradation beyond the optimal point holds universally for deep models, then early stopping is likely to be a beneficial heuristic for improving performance.
>
> Bias-fixing: In the linear model, we demonstrated the difficulty of simultaneously optimizing two distinct components (the bias and the decision boundary's orientation) via SD (Figure 4B). It is natural to expect that this difficulty would persist in more complex models. Therefore, fixing the bias term is likely a better strategy than leaving it unconstrained, even in large-scale settings. Furthermore, even if this is not directly applicable, the concepts from our analysis can inspire more sophisticated heuristics for larger models. As mentioned in the manuscript (lines 344-345), exploring strategies like the sequential optimization of different parameter groups via SD is a promising future direction.

---

> > ### Comment · Reviewer_F2RL · 2025-08-05
> >
> > I would like to thank the authors for addressing my concerns. I still believe that training the full ResNet models on CIFAR datasets should be doable. Including the additional experiments on these could further enhance the paper. Nevertheless, I think the theoretical insights are valuable.

---

### Author Response · Authors · 2025-08-07
**Follow-up to our Rebuttal**

Thank you again for your valuable feedback on our manuscript.
In our recently submitted rebuttal, we have provided answers to the questions you raised and included detailed explanations regarding your comments.
We would be very grateful if you could take a moment to review our response. We are, of course, happy to provide any further clarification if needed.

---

### Note · Authors · 2025-08-13

We thank the reviewers for their thoughtful comments. We will polish the manuscript for the camera-ready version. In particular:

1. On clarifying our core contribution (regarding the comment by qzrp)
- Regarding the benefits of SD: The reviewer pointed out that an optimal classifier can be obtained without SD, and on that basis, commented that “the current result is limited in scope”. However, our key argument is that SD can, in fact, surpass the performance of the Bayes-optimal classifier. In the camera-ready version, we will revise our explanations and add theoretical emphasis to make this contribution unambiguous.
- Regarding our hyperparameter setting: We also appreciate the comment on our choice of the hyperparameter $\lambda_t\to\infty$ in the analysis of early-stopping heuristics. This choice was found by numerical optimization without assuming prior knowledge on the hyperparameter. Also, this setting is a natural choice for the linear models we analyze. However, we acknowledge that it is not guaranteed in more general models. We will clarify the scope and rationale of this choice more explicitly in the camera-ready version.

2. On the Scope of Numerical Experiments\
We appreciate the suggestions to validate our theory on more complex settings, such as training all layers of ResNet or using the ImageNet. We agree that these are important directions for future work to demonstrate the broader applicability.
However, we believe that our current experiments serve as a standard benchmark to validate our theoretical insights. Furthermore, our core contribution is providing theoretical principles for phenomena that are not easily verifiable through large-scale experiments, which can be computationally infeasible even for single-hyperparameter black-box optimization.


3. On extensions of the theoretical model\
The reviewers suggested several valuable extensions, like multi-class classification, structured noise, and deeper models. We agree that these represent theoretically interesting avenues for future research.
However, these extensions could require new analysis and potentially deviate from the main focus of this paper: clarifying the foundational role of SD. Therefore, to encourage follow-up studies based on our work, we plan to summarize these promising possibilities more formally as future work in the camera-ready version.

We believe these refinements will improve the manuscript's clarity and accessibility. Thank you for your consideration.

---

### Decision · Program_Chairs · 2025-09-17

**Decision:**

Accept (poster)

**Comment:**

This paper presents a theoretical analysis of multi-stage self-distillation (SD) for a linear classifier on noisy Gaussian mixture data. Using the replica method from statistical physics, the authors derive a precise characterization of the generalization error, allowing for an exhaustive, optimal tuning of the SD process. The main findings are that the primary benefit of SD stems from a denoising effect via hard pseudo-labels, with the most significant gains observed in moderately-sized datasets. The paper also proposes two practical heuristics: early stopping to prevent performance degradation over many stages, and bias-fixing to handle label imbalance. The primary strength of this work lies in its rigorous analytical framework, which provides a rare, microscopic view of a complex learning dynamic. This theoretical clarity leads to principled insights that are otherwise difficult to obtain from purely empirical studies, where exhaustive hyperparameter tuning is often computationally infeasible. The main weakness, acknowledged by the authors and all reviewers, is the gap between the simplified theoretical model and the complex, deep neural networks where SD is often applied in practice.

The paper initiated a helpful discussion among the reviewers and authors. Reviewers raised several important points, including the need for broader experimental validation on more complex models, the non-rigorous nature of the replica-based "theorems", and the restrictive assumptions of the theoretical model. The authors provided a thorough rebuttal, clarifying the scope of their work and agreeing to several key revisions for the final version. They convincingly argued that their experiments serve as a valuable sanity check for their theoretical insights and promised to adjust terminology to more accurately reflect the nature of their theoretical results. One reviewer (qzrp) raised a substantive concern that the benefits of multi-stage SD might be unnecessary if a single-stage Bayes-optimal classifier is attainable. In response, the authors clarified that a key finding of their work is that multi-stage SD can surpass the performance of this single-stage baseline, a point supported by their theoretical results.

My recommendation is to accept this paper. The primary reason is that the work provides a clear, precise, and insightful theoretical treatment of a widely used but poorly understood technique. While the model is simplified, the analysis yields valuable conceptual clarity on why and when self-distillation is effective, particularly regarding its denoising properties. The authors' response to the reviews was constructive and addressed the main concerns adequately. While the reservations about the model's simplicity are valid, the consensus is that the new theoretical perspective and the practical heuristics that emerge from it represent a solid contribution to the field. This work provides a strong foundation that can guide both future theoretical analysis and practical applications of self-distillation. Acceptance is contingent on the authors incorporating their promised changes, particularly regarding the terminology of their theoretical results and the addition of supporting evidence for their claims, into the camera-ready version.